JCB Journal of Cell Biology

# Clathrin coats partially preassemble and subsequently bend during endocytosis

Markus Mund[1,2]*, Aline Tschanz[1,3]*, Yu-Le Wu[1,3], Felix Frey[4], Johanna L. Mehl[1], Marko Kaksonen[2,5], Ori Avinoam[1,6], Ulrich S. Schwarz[7,8], and Jonas Ries[1]

**Eukaryotic cells use clathrin-mediated endocytosis to take up a large range of extracellular cargo. During endocytosis, a clathrin coat forms on the plasma membrane, but it remains controversial when and how it is remodeled into a spherical vesicle. Here, we use 3D superresolution microscopy to determine the precise geometry of the clathrin coat at large numbers of endocytic sites. Through pseudo-temporal sorting, we determine the average trajectory of clathrin remodeling during endocytosis. We find that clathrin coats assemble first on flat membranes to 50% of the coat area before they become rapidly and continuously bent, and this mechanism is confirmed in three cell lines. We introduce the cooperative curvature model, which is based on positive feedback for curvature generation. It accurately describes the measured shapes and dynamics of the clathrin coat and could represent a general mechanism for clathrin coat remodeling on the plasma membrane.**

## Introduction

Endocytosis is an essential function of eukaryotic cells to internalize molecules from their surface. The major endocytic pathway is clathrin-mediated endocytosis (CME), which supports the uptake of many diverse cargos including nutrients, signaling molecules, membrane proteins, and pathogens. During CME, a dense coat of proteins self-assembles on the inner leaflet of the plasma membrane. The membrane itself is bent into an Ω-shaped invagination that is eventually pinched off to form a coated vesicle, completing endocytosis (Kaksonen and Roux, 2018).

The major component of the coat is clathrin triskelia, which comprises three heavy and three light chains (Fotin et al., 2004). When triskelia are bound to the plasma membrane by adaptor proteins, they form ordered clathrin lattices. The structural flexibility of triskelia allows these lattices to adapt variable ratios of predominantly pentagonal and hexagonal faces, forming both flat and curved geometries.

Both geometries have been observed in vivo and in vitro, and their structure has been well characterized in vitro from a structural biology perspective (Cheng et al., 2007; Dannhauser and Ungewickell, 2012; Fotin et al., 2004; Heuser and Kirchhausen, 1985; Morris et al., 2019; Pearse, 1976; Smith et al., 1998; Takei et al., 1998; Ungewickell and Branton, 1981). However, it remains elusive how clathrin coat formation and membrane bending are temporally and causally related during endocytosis in cells.

In EM micrographs, it was observed early on that clathrin lattices can assume many different curvatures in cells (Heuser, 1980). Since then, two main models of clathrin coat formation during endocytosis have been put forward. In the constant area model (CAM), clathrin grows to its final surface area as a flat coat, which then becomes continuously more curved until vesicle formation is complete. This model assumes that all the differently curved clathrin structures are endocytic intermediates. Early observations suggested the presence of pentagonal and hexagonal faces in isolated coated vesicles (Kanaseki and Kadota, 1969). Combined with the observation of flat lattices being enriched in hexagons, it was suggested that the integration of at least 12 pentagonal faces is a prerequisite for the formation of a spherical structure (Heuser, 1980). However, this would require extensive lattice remodeling, which was deemed thermodynamically and structurally unfavorable and thus unlikely to occur (Kirchhausen, 1993). The constant curvature model (CCM) was therefore formulated, which assumes that flat clathrin structures are not endocytic precursors. Instead, it was proposed

¹Cell Biology and Biophysics, European Molecular Biology Laboratory, Heidelberg, Germany;   ²Department of Biochemistry, University of Geneva, Geneva, Switzerland;   ³Candidate for Joint PhD Programme of EMBL and University of Heidelberg, Heidelberg, Germany;   ⁴Kavli Institute of Nanoscience, Department of Bionanoscience, Delft University of Technology, Delft, Netherlands;   ⁵NCCR Chemical Biology, University of Geneva, Geneva, Switzerland;   ⁶Department of Biomolecular Sciences, Weizmann Institute of Science, Rehovot, Israel;   ⁷Institute for Theoretical Physics and Bioquant, Heidelberg University, Heidelberg, Germany;   ⁸Bioquant, Heidelberg University, Heidelberg, Germany.

*M. Mund and A. Tschanz contributed equally to this paper.   Correspondence to Jonas Ries: Jonas.ries@embl.de

J.L. Mehl's current affiliation is ETH Zurich, Switzerland.   F. Frey's current affiliation is Institute of Science and Technology Austria, Klosterneuburg, Austria.

that the endocytic clathrin coat assumes its final curvature, i.e., the curvature of the vesicle, directly from the start, while continuously growing in surface area over time.

The constant curvature model had been prevalent in the field but was recently challenged by reports that flat clathrin coats can indeed change curvature during endocytosis (Avinoam et al., 2015; Bucher et al., 2018; Scott et al., 2018). This has once again boosted interest in clathrin remodeling (Chen and Schmid, 2020; Kaksonen and Roux, 2018; Sochacki and Taraska, 2018). Recent years have seen numerous studies based on diverse methods that did not converge on a common result. Reports supported either the constant area model (Avinoam et al., 2015; Sochacki et al., 2021), the constant curvature model (Willy et al., 2021), a combination of both models (Bucher et al., 2018; Tagiltsev et al., 2021; Yoshida et al., 2018), or the simultaneous existence of both models within cells (Scott et al., 2018). To conclusively understand this complex process, it would be desirable to directly visualize the 3D nanoscale structure of the endocytic clathrin coat in living cells. Unfortunately, however, to date, no method offers the necessary spatial and temporal resolution to do that. Here, we aim to circumvent this limitation by densely sampling the entire endocytic process in fixed cells, subsequently reconstructing the dynamic information.

We developed a superresolution microscopy approach to quantitatively study the 3D clathrin coat architecture at endocytic sites. We used a novel model fitting framework to extract geometric parameters that allowed us to sort static images of clathrin lattices according to their progression along the endocytic timeline. The inferred endocytic dynamics allowed us to reconstruct the stereotypic remodeling of clathrin during endocytosis at the nanoscale.

In summary, we found that a clathrin coat first partially assembles on a flat membrane and then simultaneously grows in surface area and coat curvature. While initial bending occurs rapidly, it later slows down and endocytic sites are eventually paused in a state of high curvature before vesicle scission. This trend is conserved across cell lines, suggesting it is a common attribute that is not affected by cell type. Based on this data, we developed a new kinetic growth model, the cooperative curvature model (CoopCM). It describes coat area growth as the addition of clathrin to the lattice edge with a constant rate and assumes that the curvature of the coat increases toward a preferred curvature, driven by the cooperative interplay of clathrin triskelia within the lattice. The CoopCM predicts a fast initial curvature increase that is slowed down progressively as the coat becomes spherical and shows excellent agreement with experimental data.

## Results

### Quantitative 3D superresolution imaging of clathrin structures

Here, we used 3D single-molecule localization microscopy (SMLM) to systematically determine the precise geometry of individual clathrin-coated structures at the plasma membrane. For this, we optimized the sample preparation to label clathrin at endocytic sites as densely as possible using indirect immuno-fluorescence with polyclonal antibodies against clathrin light and heavy chains, which was crucial to achieve high-quality SMLM (Mund and Ries, 2020). We then localized sparsely activated single fluorophores by fitting an experimentally derived model of the astigmatic point-spread function using a method we developed previously (Li et al., 2018). This improved the resolution to about 10 nm in x/y and 30 nm in z (based on modal values of the localization precision at 3.9 nm in x/y and 12.5 nm in z; see Materials and methods) and reduced typically observed image distortions. The achieved image quality allowed us to clearly visualize and quantify the 3D clathrin coat shape at different stages of endocytic site maturation (Fig. 1, A–C).

The large majority of sites were single structures that were well-isolated from each other and exhibited great structural diversity. In addition, we noted several clusters of closely juxtaposed endocytic sites (Fig. S1 A). In the isolated sites, we observe a variety of shapes including flat, curved, dome-shaped, and spherical structures of different sizes (Fig. 1 C), indicating that endocytosis has been arrested by fixation at different time points during endocytic progression.

To quantify the size and shape of individual clathrin coats, we used LocMoFit, a computational pipeline based on a maximum-likelihood model fitting framework that we developed recently (Wu et al., 2023). This framework directly fits 3D geometric models to the 3D point cloud of localizations (Fig. 1, D and E) instead of rendered images and thereby harnesses the full information contained in the SMLM data, including, for instance, the localization precision, rather than just spatial coordinates.

We describe the clathrin coat as a spherical cap that is defined by a radius $r$ and a closing angle $\theta$ (Fig. 1 E). Our model also accounts for coat thickness, antibody size, and blurring due to the localization precision (Materials and methods). Hence, it describes both shallow and highly curved structures equally well. Moreover, because $\theta$ increases during endocytosis from 0° at flat sites to 180° at complete vesicles, we could use this parameter to sort the individual images according to endocytic progress (Fig. 1 F).

### The clathrin coat grows in area and becomes more curved during endocytosis

We first imaged immunostained clathrin in chemically fixed SK-MEL-2 cells, where we focused on the bottom plasma membrane that was densely covered by endocytic sites. These cells have been extensively studied and are established model cells for clathrin-mediated endocytosis with well-characterized endocytic dynamics (Doyon et al., 2011; Aguet et al., 2013; Avinoam et al., 2015; Kaplan et al., 2022). Using the 3D model fitting pipeline, we determined radius $R$, closing angle $\theta$, position, and rotation of 1,798 endocytic sites from 13 cells (Fig. 2 A) with high accuracy (Fig. S2). We found that two structural intermediates were enriched, while others were rare and only a small fraction of sites were completely flat (Fig. 2 B). Slightly curved sites with $\theta \approx 70°$ and strongly curved sites with $\theta \approx 130°$ were enriched, indicating that those structural states are more frequently found in cells. We only rarely obtained sites with $\theta \approx 180°$, which would be expected for complete spherical coats, even though fully formed vesicles are found in our data (Fig. S1 B). This indicates that at the time point of scission, the clathrin coat of nascent

Figure 1. **3D single-molecule localization microscopy (SMLM) of clathrin-coated structures. (A)** 3D SMLM image of clathrin immunolabeled with AF647 at the bottom membrane of a SK-MEL-2 cell. **(B)** Enlarged view of the region indicated in A. **(C)** All structures found in B are shown in top view (xy) and as 50 nm-thick z-slices (xz), and orientation of slice is indicated by a dotted line. Scale bars are 10 µm (A); 1 µm (B); and 100 nm (C). **(D)** Geometric analysis pipeline. All clathrin coats are segmented, and their localizations are directly fitted with a spherical cap model using the fitting framework LocMoFit (Wu et al., 2023). **(E)** In LocMoFit, individual clathrin coats are parametrized by their size (radius $R$), closing angle ($\theta$), position ($x_0,y_0,z_0$), and 3D orientation. **(F)** Using $\theta$ as a proxy for endocytic progression, the relative endocytic time point for each structure can be determined.

vesicles is still incomplete at the neck or that the kinetics of scission are too transient for detection. Deeply invaginated clathrin-coated pits could further be sheared off during sample preparation, thus evading our detection. Curvatures $H = 1/R$ ranged from 0 to 0.022 nm$^{-1}$ with a median of 0.011 nm$^{-1}$, corresponding to a median radius of 87 nm, and the surface areas $A$ ranged from 9,000 to 140,000 nm$^2$ with a median of 54,000 nm$^2$. These values agree well with previous measurements of the vesicular coat using EM (Avinoam et al., 2015), where a median curvature of 0.015 nm$^{-1}$ and median surface area of 54,500 nm$^2$ was measured.

We then wanted to understand how the clathrin coat geometry changes during endocytosis. To this end, we used the closing angle parameter $\theta$ to sort endocytic sites relative to each other in time. Irrespective of whether endocytosis follows a constant curvature or constant area model, $\theta$ monotonically increases from early to late endocytic time points (Fig. 2 C), and can thus be used as an unbiased proxy for endocytic progression (Avinoam et al., 2015).

The curvature $H$ was strongly correlated with $\theta$, indicating that coat curvature increases continuously during endocytosis (Fig. 2 D). Similarly, the surface area $A$ increased from 32,000 nm$^2$ (median of 5% of sites with smallest $\theta$) to 50,000 nm$^2$ (median of 5% of sites with highest $\theta$). The projected area $A_p$ decreased from 31,000 to 13,000 nm$^2$ (median of 5% of sites with lowest and highest $\theta$ respectively), which is in close agreement with previous EM measurements (Bucher et al., 2018). It is readily obvious that our data are incompatible with the constant

curvature model, as the curvature is not constant, but increases monotonically with $\theta$. Just as clearly, our data do not support the constant area model because the coat surface also increases during endocytosis.

Almost all data points are part of one continuous point cloud (Fig. 2 D), indicating a continuous transition between structural states during endocytosis. We noticed an additional small, disconnected set of data points representing 8.5% of all sites that correspond to endocytic sites with curvatures above 0.016 nm$^{-1}$ and $\theta$ of 80°–180° (Fig. 2 D, and example structures in Fig. S3, A and B). In a control experiment to check whether these are endocytic structures, we selectively analyzed clathrin structures that colocalized with AP-2, a bona fide marker for CME. We did not observe AP-2 in any of the disconnected sites, from which we conclude that they indeed do not belong to CME. These small structures could represent coated vesicles from the trans Golgi (Fig. S3).

**The cooperative curvature model of clathrin coat remodeling**

Our data clearly showed that clathrin coats grow and become more curved as the plasma membrane gets bent to produce a vesicle. Since the data quantitatively describes all 3D geometries that the endocytic clathrin coat assumes, it allowed us to move toward a mathematical model of clathrin coat formation during endocytosis.

Here, we introduce the Cooperative Curvature Model (CoopCM), which describes coat growth based on known structural and dynamical properties of clathrin coats (Fig. 3 A). First,

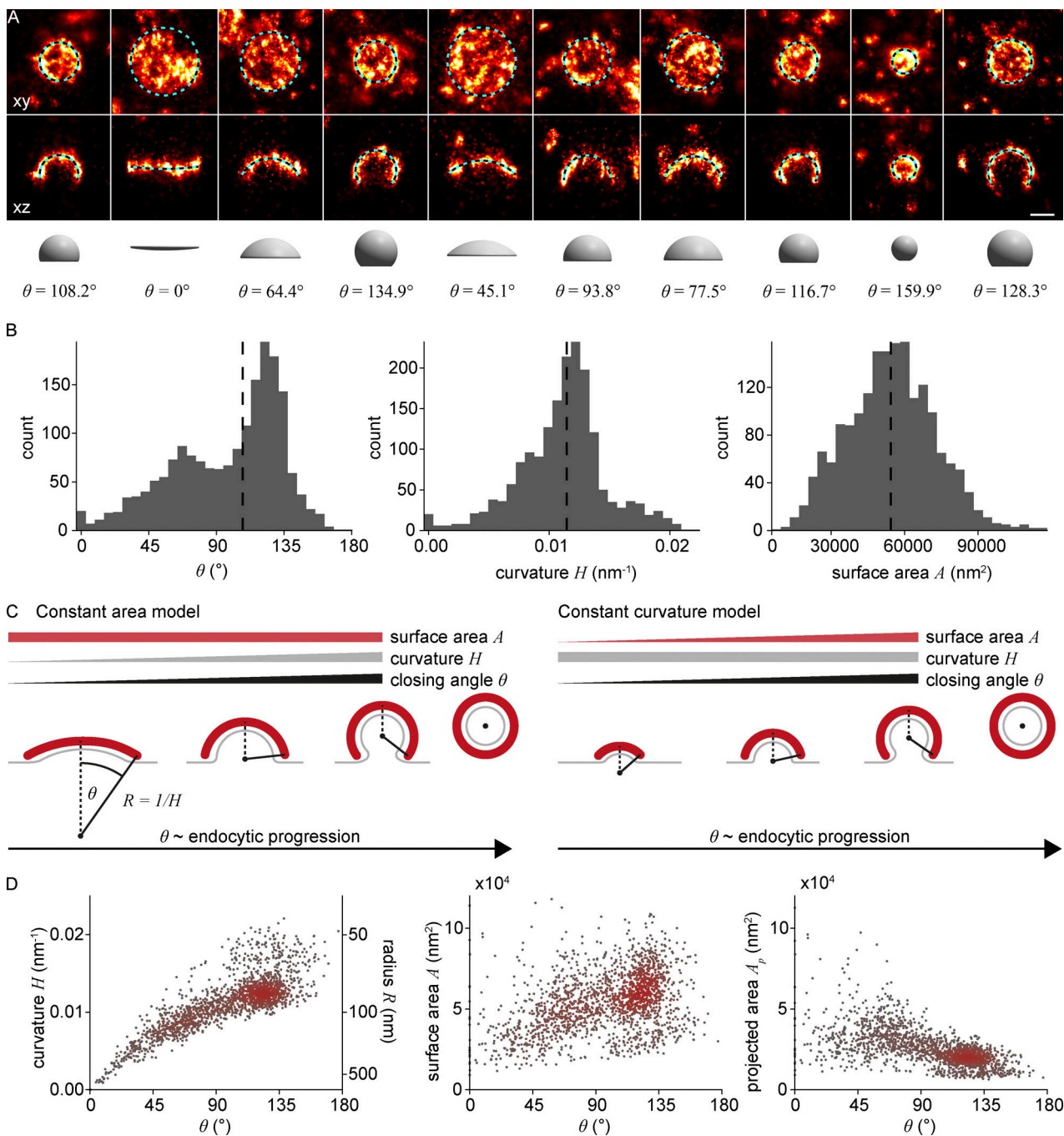

**Figure 2. Quantitative analysis of clathrin-coated structures in SK-MEL-2 cells. (A)** Clathrin coat geometry is quantified using LocMoFit. The fitted spherical cap model is drawn as a circle with its outer radius (top row, xy), as cross-sectional profile (50 nm xz slices, middle row), and as surface with the corresponding closing angle θ (bottom row). Scale bar, 100 nm. **(B)** Distributions of closing angle θ (median = 108°), curvature H (median = 0.011 nm[-1]), and surface area A (median = 54,000 nm[2]) of endocytic sites in SK-MEL-2 cells (n = 1,798 sites, N = 13 cells) as determined from the model fit. **(C)** Two previously proposed mechanistic models for clathrin coat assembly during endocytosis (for details, see text). In both scenarios, θ increases monotonically and is thus a proxy for endocytic progression. **(D)** Development of curvature, surface area, and projected area of the clathrin coat during endocytosis. Color indicates point density.

we assume that the clathrin coat starts growing on a flat membrane. While triskelia have been shown to exchange dynamically during endocytosis (Avinoam et al., 2015), net growth of the area $A$ occurs via the addition of triskelia at the lattice edge $\varepsilon$ with a constant growth rate $k_{on}$ (Eq. 1).

$$\frac{d}{dt}A = \dot{A} = k_{on}\varepsilon. \tag{1}$$

Moreover, we assume that the intrinsic pucker angle of individual triskelia and their interactions in a lattice together translate into an overall preferred curvature $H_0$ of the clathrin

**Figure 3. Model for clathrin coat growth. (A)** Schematic of the cooperative curvature model, where clathrin lattices grow by the addition of triskelia to the edge at a constant growth rate $k_{on}$. Curvature generation is driven toward a preferred curvature, ultimately creating a spherical vesicle. **(B)** Distinct clathrin growth models and rolling median (window width = 82 sites) fitted to curvature over θ. **(C and D)** The resulting fitting parameters are then used to map the same models also over (C) surface area and (D) edge length. ($n$ = 1,645 sites, $N$ = 13 cells).

coat as a whole. However, the initially flat assembly suggests that this preferred curvature cannot be realized in an immature lattice. We hypothesize that cooperative processes in the coat are required to increase curvature. Coat curvature $H$ then increases asymptotically toward $H_0$ at an initial rate γ, slowing down toward zero when the preferred curvature is reached (Eq. 2).

$$\frac{d}{d\theta}H = \gamma\left(1 - \frac{H^2}{H_0^2}\right). \quad (2)$$

Choosing a quadratic dependence of the rate of curvature change on curvature is a simple way to represent cooperativity in the lattice, which recently has been demonstrated in experiments (Sochacki et al., 2021; Zeno et al., 2021) and fits our data more accurately than the linear relationship, which would correspond to a less cooperative process (Appendix). Eq. 2 can be solved analytically to yield an expression of the curvature $H$ depending on θ (Eq. 3), which can then be fitted to our data.

$$H(\theta) = H_0 \tanh\left(\frac{\gamma\theta}{H_0}\right). \quad (3)$$

Analogous expressions for other geometric parameters like surface area can be derived straightforwardly.

The fit of this CoopCM to the data of curvature $H$ in relation to the closing angle θ shows excellent agreement (Fig. 3 B). In comparison, the CAM fitted the curvature data slightly worse than the CoopCM, and the CCM agreed poorly with the data. The improved fit of the CoopCM compared with the CAM or CCM is not a result of the additional fitting parameter, as we showed using the Bayesian Information Criterion (see Appendix).

From the curvature fits, we calculated the corresponding curves for surface area (Fig. 3 C) and the edge length of the clathrin coat (Fig. 3 D). The edge length decreases monotonically and approaches zero at θ = 180°, thereby stopping growth according to Eq. 1. Both graphs again highlight the very close agreement of the model prediction with the experimental data. From the fit, we determined that invagination occurs when about half of the total coat area has grown ($A_0$ = 0.51), and that the preferred curvature of a clathrin coat is $H_0$ = 0.014 nm⁻¹, corresponding to a radius of $R_0$ = 72 nm. The model yields nearly identical parameters when the surface area or edge length is fitted instead of curvature (Table 1), highlighting its robustness.

We then decided to test if the observed mode of clathrin remodeling is specific for SK-MEL-2 cells. For this, we analyzed the endocytic clathrin ultrastructure also in U2OS cells and 3T3 mouse fibroblasts (Fig. S4; and Tables 2 and 3). In these cell lines, just like in SK-MEL-2 cells, the curvature as well as the surface area continuously increase throughout endocytosis. We observed that the preferred curvature of the clathrin coat is smaller in U2OS ($H_0$ = 0.012 nm⁻¹, $R_0$ = 85 nm) and 3T3 cells ($H_0$ = 0.011 nm⁻¹, $R_0$ = 89.7 nm) compared with SK-MEL-2 ($H_0$ = 0.014 nm⁻¹, $R_0$ = 72 nm). This suggests that the average size of vesicles formed in CME is cell-line specific. The fraction of the surface area acquired on the flat membrane is very similar for all three cell lines, with U2OS-derived sites initiating curvature at $A_0$ = 0.52 and 3T3 sites at $A_0$ = 0.45 (Tables 2 and 3). Taken together, we have shown in several cell lines that clathrin coats neither grow via constant curvature or constant area pathways, but rather first grow flat and then acquire curvature and more surface area simultaneously, with a nonlinear mode of curvature generation.

## Temporal reconstruction of structural dynamics in endocytosis

We systematically segmented all endocytic sites in the super-resolution images and thereby obtained a large dataset where all endocytic time points were sampled homogeneously. The distribution of structural states within the dataset is thus representative of their lifetime, with long-lived, stable curvature states being overrepresented in the data compared to transient states. This opens up the possibility to reconstruct the temporal progression of clathrin remodeling during endocytosis and to ask whether clathrin coats acquire their curvatures at a constant rate or if there are certain curvature transitions that occur more rapidly than others.

For this, we sorted all endocytic sites by θ. The rank of an endocytic site thus corresponds to its pseudotime, which describes its relative time point between 0 and 1 along the endocytic trajectory (Fig. 4 A).

As our model (Eq. 1) describes the dynamic growth of the clathrin coat, we can solve it to derive an expression for θ over time $t$ (Eq. 4), where the coat starts out flat and then starts to generate curvature, increasing θ over time.

**Table 1. Summary of clathrin coat growth model fits in SK-MEL-2.**

| | | Curvature H(θ) | Surface area A(θ) | Edge length Ɛ(θ) | Projected area $A_p$(θ) |
|---|---|---|---|---|---|
| **CAM** | **A** | $58{,}300 \pm 400$ nm$^2$ | $56{,}600 \pm 400$ nm$^2$ | $49{,}400 \pm 500$ nm$^2$ | $48{,}800 \pm 500$ nm$^2$ |
| **CCM** | **R** | $94.5 \pm 0.7$ nm | $84.8 \pm 0.4$ nm | $97.3 \pm 0.8$ nm | $92.8 \pm 0.7$ nm |
| **CoopCM** | **γ** | $(9.4 \pm 0.1)\ 10^{-3}$ nm$^{-1}$ | $(9.0 \pm 0.1)\ 10^{-3}$ nm$^{-1}$ | $(9.4 \pm 0.1)\ 10^{-3}$ nm$^{-1}$ | $(9.1 \pm 0.1)\ 10^{-3}$ nm$^{-1}$ |
| | **$H_0$** | $(13.9 \pm 0.1)\ 10^{-3}$ nm$^{-1}$ | $(13.8 \pm 0.1)\ 10^{-3}$ nm$^{-1}$ | $(13.4 \pm 0.1)\ 10^{-3}$ nm$^{-1}$ | $(13.6 \pm 0.1)\ 10^{-3}$ nm$^{-1}$ |
| | **$R_0$** | $72.0 \pm 0.6$ nm | $72.2 \pm 0.6$ nm | $74 \pm 0.8$ nm | $73.4 \pm 0.9$ nm |
| | **$A_0$** | 0.51 | 0.55 | 0.49 | 0.52 |
| | **$k_{on}$** | 78.1 nm$\tilde{s}^{-1}$ | 72.2 nm$\tilde{s}^{-1}$ | 83.4 nm$\tilde{s}^{-1}$ | 78 nm$\tilde{s}^{-1}$ |

Fitted parameter values for the constant area model (CAM), the constant curvature model (CCM) and the cooperative curvature model (CoopCM) when fitting curvature $H(\theta)$, surface area $A(\theta)$, edge length $Ɛ(\theta)$, and projected area $A_p(\theta)$. A: Surface area fitted with CAM; R: Radius fitted with CCM; γ: Constant rate of curvature increase fitted with CoopCM; $H_0$: Preferred curvature of the clathrin coat fitted with CoopCM; $R_0$: preferred radius of the clathrin coat fitted with CoopCM; $A_0$: Fraction of surface area growing as a flat lattice before curvature initiation, defined as $A_0 = A(\theta = 0.01)/A(\theta = 180°)$; $k_{on}$: local growth rate obtained from $\theta(t)$ fitted with CoopCM and measured in nm per pseudotime units $\tilde{s}^{-1}$. $N$ = 13 cells; $n$ = 1,645 sites.

$$\theta(t) = \sqrt{\frac{24\gamma k_{on}}{8\gamma^2 H_0^{-2} - 1}t}. \qquad (4)$$

The square root dependence of $\theta$ on time $t$ reflects the slowing down of curvature generation as the clathrin coat approaches its preferred curvature. This expression fits the pseudotime-resolved data remarkably well (Fig. 4 B). Consistent with our previous reasoning, a linear model did not agree well with the data (Appendix), since it gives a linear invagination speed for small $t$, emphasizing the validity of our cooperative curvature model, which leads to the characteristic square root dependence. The data slightly deviates from the model in the early phase and more notably in the very end for pseudotimes close to 1. This potentially indicates that clathrin geometry is influenced by other factors besides the coat itself close to vesicle scission.

Since this pseudotime-resolved dataset was generated from a large number of endocytic sites in many cells, we effectively generated the average trajectories of how curvature, surface area, projected area, and lattice edge change during endocytosis in SK-MEL-2 cells (Fig. 4, C–F). We observe comparatively few flat clathrin coats. This shows that clathrin lattices are only transiently flat at the beginning of endocytosis and might represent an energetically unfavorable conformation (Fig. 4 B). A fast transition from a flat to a curved structure could further be mediated independently of clathrin coat (Zhao et al., 2017), e.g., by the action of BAR domain proteins (Henne et al., 2010) or at the base of filopodial projections. We further find comparatively many structures with a curvature of $\theta \approx 70°$ and $\theta \approx 130°$, indicating more long-lived endocytic stages. As the surface area is constantly increasing over pseudotime, we assume that the enrichment at $\theta \approx 130°$ represents a stalling point, where vesicles are formed by the addition of the final clathrin triskelia and potentially other factors that enable the recruitment and mechanic function of dynamin during vesicle scission. Similarly, the enrichment at $\theta \approx 70°$ could be indicative of further recruitment of regulatory components, potentially supporting a previously suggested endocytic checkpoint (Loerke et al., 2009).

**Table 2. Summary of clathrin coat growth model fits in U2OS.**

| | | Curvature H(θ) | Surface area A(θ) | Edge length Ɛ(θ) | Projected area $A_p$(θ) |
|---|---|---|---|---|---|
| **CAM** | **A** | $80{,}800 \pm 1{,}600$ nm$^2$ | $76{,}900 \pm 1{,}700$ nm$^2$ | $65{,}900 \pm 1{,}600$ nm$^2$ | $65{,}400 \pm 1{,}600$ nm$^2$ |
| **CCM** | **R** | $119.1 \pm 2.5$ nm | $101.6 \pm 1.5$ nm | $122.8 \pm 2.7$ nm | $115 \pm 2.4$ nm |
| **CoopCM** | **γ** | $(7.9 \pm 0.2)\ 10^{-3}$ nm$^{-1}$ | $(7.7 \pm 0.2)\ 10^{-3}$ nm$^{-1}$ | $(8.1 \pm 0.2)\ 10^{-3}$ nm$^{-1}$ | $(7.9 \pm 0.2)\ 10^{-3}$ nm$^{-1}$ |
| | **$H_0$** | $(11.8 \pm 0.2)\ 10^{-3}$ nm$^{-1}$ | $(11.5 \pm 0.2)\ 10^{-3}$ nm$^{-1}$ | $(11.0 \pm 0.3)\ 10^{-3}$ nm$^{-1}$ | $(11.1 \pm 0.3)\ 10^{-3}$ nm$^{-1}$ |
| | **$R_0$** | $85.0 \pm 1.8$ nm | $87.1 \pm 1.7$ nm | $90.5 \pm 2.7$ nm | $90.2 \pm 2.6$ nm |
| | **$A_0$** | 0.52 | 0.52 | 0.45 | 0.47 |
| | **$k_{on}$** | 89.3 nm$\tilde{s}^{-1}$ | 91.5 nm$\tilde{s}^{-1}$ | 110.3 nm$\tilde{s}^{-1}$ | 105.5 nm$\tilde{s}^{-1}$ |

Fitted parameter values for the constant area model (CAM), the constant curvature model (CCM), and the cooperative curvature model (CoopCM) when fitting curvature $H(\theta)$, surface area $A(\theta)$, edge length $Ɛ(\theta)$, and projected area $A_p(\theta)$. A: Surface area fitted with CAM; R: Radius fitted with CCM; γ: Constant rate of curvature increase fitted with CoopCM; $H_0$: Preferred curvature of the clathrin coat fitted with CoopCM; $R_0$: preferred radius of the clathrin coat fitted with CoopCM; $A_0$: Fraction of surface area growing as a flat lattice before curvature initiation, defined as $A_0 = A(\theta = 0.01)/A(\theta = 180°)$; $k_{on}$: local growth rate obtained from $\theta(t)$ fitted with CoopCM and measured in nm per pseudotime units $\tilde{s}^{-1}$. $N$ = 3 cells; $n$ = 241 sites.

**Table 3.  Summary of clathrin coat growth model fits in 3T3 mouse fibroblasts.**

| | | Curvature H(θ) | Surface area A(θ) | Edge length Ɛ(θ) | Projected area $A_p$(θ) |
|---|---|---|---|---|---|
| **CAM** | **A** | $90{,}100 \pm 1{,}000$ nm$^2$ | $83{,}700 \pm 1{,}000$ nm$^2$ | $65{,}800 \pm 1{,}000$ nm$^2$ | $66{,}200 \pm 1{,}000$ nm$^2$ |
| **CCM** | **R** | $115.8 \pm 1.5$ nm | $98.3 \pm 0.7$ nm | $121.1 \pm 2$ nm | $108.5 \pm 1.5$ nm |
| **CoopCM** | **γ** | $(8.2 \pm 0.1)\ 10^{-3}$ nm$^{-1}$ | $(7.8 \pm 0.1)\ 10^{-3}$ nm$^{-1}$ | $(7.9 \pm 0.1)\ 10^{-3}$ nm$^{-1}$ | $(7.8 \pm 0.1)\ 10^{-3}$ nm$^{-1}$ |
| | **$H_0$** | $(11.3 \pm 0.1)\ 10^{-3}$ nm$^{-1}$ | $(11.2 \pm 0.1)\ 10^{-3}$ nm$^{-1}$ | $(11.2 \pm 0.2)\ 10^{-3}$ nm$^{-1}$ | $(11.2 \pm 0.2)\ 10^{-3}$ nm$^{-1}$ |
| | **$R_0$** | $88.8 \pm 0.7$ nm | $89.3 \pm 0.7$ nm | $89.1 \pm 1.4$ nm | $89.5 \pm 1.3$ nm |
| | **$A_0$** | 0.45 | 0.49 | 0.47 | 0.49 |
| | **$k_{on}$** | $128.4$ nm$\tilde{s}^{-1}$ | $120.8$ nm$\tilde{s}^{-1}$ | $123.9$ nm$\tilde{s}^{-1}$ | $121$ nm$\tilde{s}^{-1}$ |

Fitted parameter values for the constant area model (CAM), the constant curvature model (CCM), and the cooperative curvature model (CoopCM) when fitting curvature $H(\theta)$, surface area $A(\theta)$, edge length $\mathcal{E}(\theta)$, and projected area $A_p(\theta)$. A: Surface area fitted with CAM; R: Radius fitted with CCM; γ: Constant rate of curvature increase fitted with CoopCM; $H_0$: Preferred curvature of the clathrin coat fitted with CoopCM; $R_0$: preferred radius of the clathrin coat fitted with CoopCM; $A_0$: Fraction of surface area growing as a flat lattice before curvature initiation, defined as $A_0 = A(\theta = 0.01)/A(\theta = 180°)$; $k_{on}$: local growth rate obtained from $\theta(t)$ fitted with CoopCM and measured in nm per pseudotime units $\tilde{s}^{-1}$. N = 7 cells; n = 688 sites.

This functional interpretation is supported by the observation of similar peaks in U2OS (at $\theta \approx 60°$ and ≈130°) as well as 3T3 cells (at $\theta \approx 50°$ and ≈130°).

Within the first 10% of pseudotemporal progression, the clathrin coat rapidly acquires shallow curvature of H = 0.007 nm$^{-1}$ (R = 134 nm). It then becomes gradually more bent up to

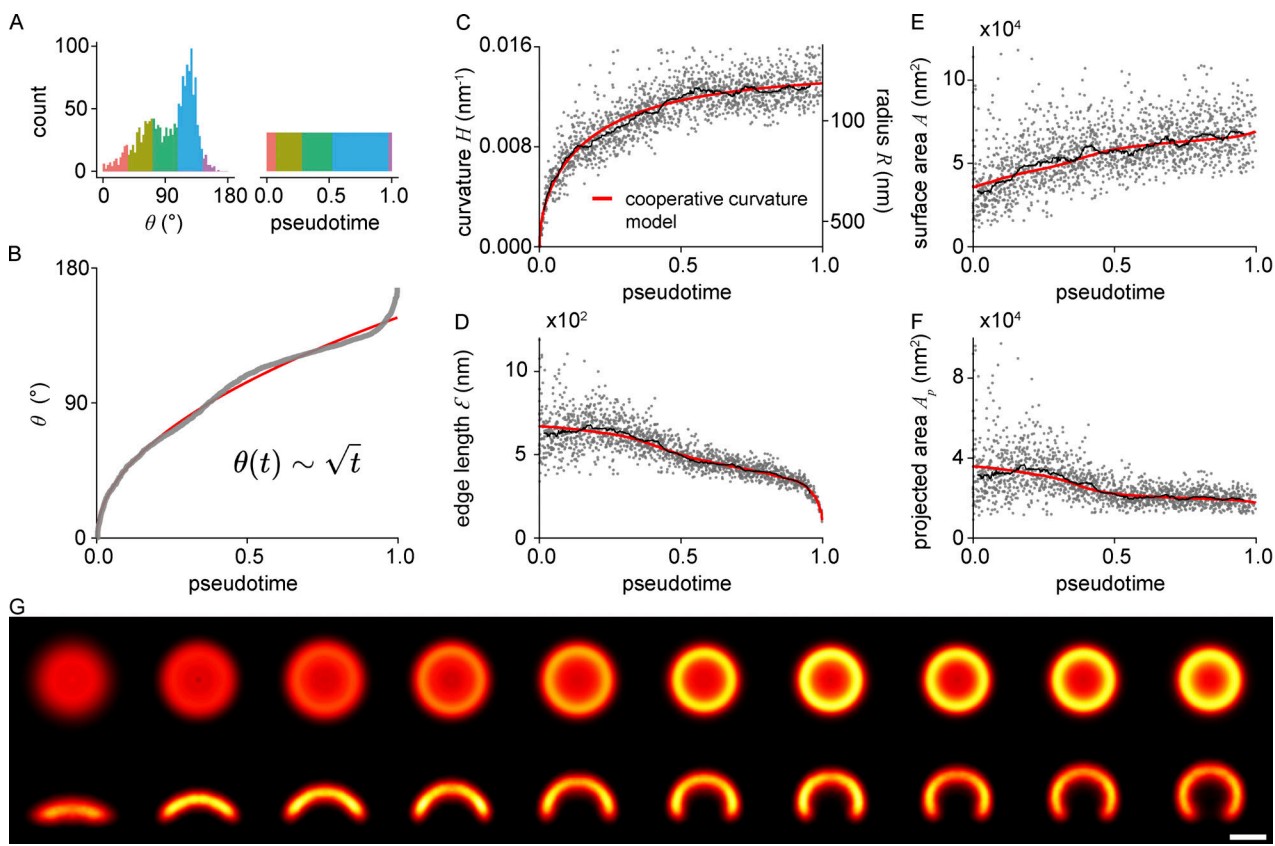

Figure 4.  **Temporal reconstruction of clathrin coat remodeling. (A)** Endocytic sites are sorted by θ to reconstruct pseudotime. Enriched θ states, for example the peak at 135°, represent long-lived states that remain for extended periods in pseudotime. Color code represents the same number of sites in both histograms. **(B)** The square-root dependence between θ and pseudotime approximated by the cooperative curvature model (red line). **(C)** Curvature generation over pseudotime is approximated by the cooperative curvature model. **(D–F)** Fit results in C were used to describe (D) edge length, (E) surface area, and (F) projected area change over pseudotime. A rolling median (window of 82 sites) is plotted alongside (black line). **(G)** Superresolution averages for distinct endocytic stages, resulting from all collected snapshots. Each bin contains the same number of snapshots of clathrin-coated structures sorted along their pseudotime (n = 163 per bin), so that all bins represent equally long pseudotime intervals. Individual sites were rescaled to the median bin radius and aligned by their center coordinates as well as rotation angles. Scale bar is 100 nm. (n = 1,645 sites, N = 13 cells).

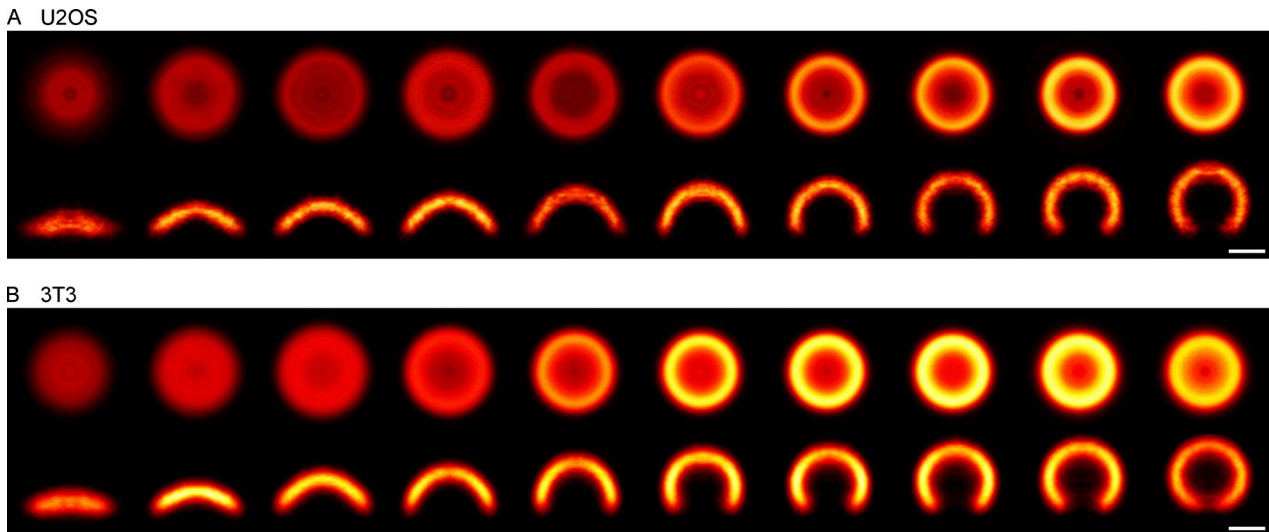

**Figure 5. Temporal reconstruction of clathrin coat remodeling across three different cell lines. (A and B)** Superresolution averages for distinct endocytic stages, resulting from all collected snapshots for (A) U2OS (*n* = 241 sites, *N* = 3 cells) and (B) 3T3 (*n* = 688 sites, *N* = 7 cells). Each bin contains the same number of snapshots of clathrin-coated structures sorted along their pseudotime ($n_{U2OS}$ = 24 per bin, $n_{3T3}$ = 68) and represent equally long pseudotime intervals.

≈60% of the endocytic timeline, when the sites reach an average curvature of 0.012 nm$^{-1}$ (*R* = 83 nm). During the last ≈40% of progression, the curvature increase almost stops, until vesicle scission occurs. Interestingly, the earliest sites in our dataset already contain ≈50% of the surface area of the final vesicles, which is also reflected in the fitting results ($A_O$ = 0.51; Fig. 4 E). This indicates that the initial assembly of the clathrin coat occurs very rapidly or is in part below our detection limit.

Finally, we made a 3D nanoscale movie, which vividly illustrates the assembly and remodeling of the clathrin coat during endocytosis from pseudotime-resolved averages of our data (Fig. 4 G). This yielded a nanoscale pseudo-temporal movie of endocytosis in SK-MEL-2 cells (Video 1).

We performed identical analyses for U2OS- and 3T3-derived clathrin sites and obtained highly similar results as for SK-MEL-2 cells, indicating that this trajectory represents a general pathway for clathrin coat remodeling during endocytosis (Fig. S4; and Tables 2 and 3). Also, these data can be represented as high-resolution averages (Fig. 5).

## Discussion

### Quantitative description of clathrin coat ultrastructure

The nature of clathrin assembly during endocytosis has become a classical question in cell biology that remains unresolved (reviewed in Chen and Schmid, 2020; Sochacki and Taraska, 2018). There have been two main competing mechanistic models of how triskelia form a coat: the constant curvature model predicts that the clathrin coat assumes its final curvature directly from the start, while continuously increasing in surface area over time. In contrast, the constant area model predicts that clathrin grows to its final surface area as a flat coat, which then becomes continuously more curved until vesicle formation is complete.

Each model is supported by a number of studies, which mostly relied on electron microscopy techniques. Among them, 3D correlative electron tomography yields exquisite 3D endocytic membrane shapes (Avinoam et al., 2015) but can be tedious to extend toward large numbers of cells. Platinum replica electron microscopy offers large fields of view and thus superb throughput, but gives limited 3D resolution (Bucher et al., 2018; Sochacki et al., 2017).

We reason that the trajectory of clathrin coat assembly during endocytosis could best be determined by systematically imaging large numbers of endocytic sites with high 3D resolution. Therefore, we used SMLM, which combines high 3D resolution with high throughput. Although lower than in electron microscopy, the resolution we achieved allowed us to resolve the precise 3D coat ultrastructure of clathrin-coated pits. Importantly, due to the molecular specificity for clathrin, we were able to segment all endocytic sites at the bottom surface of the cells in our images in an unbiased way, thus ensuring homogenous sampling of the structural variety of clathrin coats.

We applied a novel maximum-likelihood-based fitting framework that we developed recently (Wu et al., 2023), which allows fitting complex geometric models to localization point clouds. Since the fit is applied directly to the localizations, it considers all quantitative parameters of all localizations, most notably localization uncertainty. This results in the precise and reliable quantification of the underlying structures (Wu et al., 2020), even when taking linkage error through indirect immunolabeling into account (Fig. S5, A–L).

We described the clathrin coat as a spherical cap, which matched the large majority of sites very well. Additionally, in our data, we observed that some sites are asymmetric, ellipsoidal, or deformed more irregularly (Fig. S1 C). We do not currently evaluate this asymmetry. Rather, we introduced rotational symmetry during the averaging (Fig. 4 G), where we aligned

each site based on their model parameters and thus averaged out many irregularities. A deviation of the cross-sectional profile from a circle is nevertheless preserved in the averaging (Fig. S2, C–G), and in future studies, more complex geometric models will enable analyzing the structure and asymmetries of the coat in more detail.

## Flat clathrin-coated structures

In the first step of our analysis pipeline, we segmented all clathrin-coated structures in the superresolution images. We thereby aimed to achieve homogenous sampling to capture all endocytic intermediates with the same probability irrespective of time point, size, and orientation. Interestingly, the surfaces of the earliest detectable endocytic sites already contained half the surface area of sites in the latest CME stages. How do these earliest sites assemble? We cannot sort the very flat sites, which are comparatively rare in our dataset, in time using $\theta$ because their $\theta$ is close to zero. However, their relative fraction among all sites is still informative. As short-lived states are underrepresented, the relative absence of small flat sites in our data indicates that the initial assembly occurs rapidly.

In addition to the short lifetimes of small flat sites, two technical reasons could potentially contribute to their rare occurrence in our data. Due to their small size and potentially substoichiometric labeling, which is also noticeable as holes in larger structures (Fig. S1 D), they might not have sufficient immunostaining signal and thus they might be hard to be differentiated from small clusters of primary and secondary antibodies that are routinely observed in SMLM images. Additionally, the very first small clathrin assemblies might be meta-stable with mostly loosely bound triskelia, and thus might not be stabilized well by chemical fixation. However, it is important to note that very small sites are also rarely found in previously published electron microscopy datasets (Avinoam et al., 2015; Bucher et al., 2018; Sochacki et al., 2021). Since EM should not suffer from the same technical limitations, it seems likely that these states are indeed short-lived.

Generally, completely flat clathrin coats are rare in our data. Instead, we observed that clathrin coats quickly acquire curvature already early in endocytosis. This agrees with the early observation made using EM (Heuser, 1980) that even flat structures already had some small degree of curvature. Interestingly, an enrichment of pentagons was observed toward the edges of flat lattices, leading to the hypothesis that even without additional formation of pentagons these flat lattices have the built-in capacity to become curved and are not very stable. This notion is in agreement with recent demonstrations that flat clathrin coats store energy, which can drive coat bending without additional factors (Sochacki et al., 2021; Tagiltsev et al., 2021).

The importance of curvature generation for clathrin coat maturation is supported by previous studies that suggest failure to initiate significant curvature as a hallmark of abortive events (Loerke et al., 2009; Wang et al., 2020). Observed in live-cell studies, these dim and short-lived events fail to significantly increase in intensity and do not undergo scission (Loerke et al., 2009; Mettlen et al., 2009). As structures in our data set mostly contain at least shallow curvatures at minimally half the final coat area, we are likely not capturing these abortive events, and they have negligible impact on our analysis.

## The cooperative curvature model

Here, we visualized the shapes of clathrin coats with high 3D resolution. This was crucial to precisely measure the clathrin curvature throughout the endocytic timeline ranging from flat to highly curved structures. Especially shallow curvatures are hard to accurately assess with 2D imaging. Thus, our data enabled us to robustly test different growth models for clathrin during endocytosis.

The constant curvature model predicts a continuous increase in surface area, constant coat curvature, as well as a edge length that increases until a half-spherical geometry is reached and then decreases again. Our experimental data, most notably, the increase in curvature and monotonical decrease in edge length are incompatible with these predictions, thus ruling out the constant curvature model (Fig. 3, green lines).

The constant area model, in a strict interpretation, on the other hand, predicts the coat to assemble completely on a flat membrane, after which its area stays constant and only curvature increases to form a spherical vesicle. These predictions agree reasonably well with the data for curvature propagation during CME but fail to explain the monotonic increase in coat surface over time. Thus, the continuous increase of surface area that we observed here rules out the constant area model as well (Fig. 3, blue lines). Interestingly, earlier work compatible with the constant area model already suggested the presence of shallow curvatures in even the flattest clathrin-coated structures (Avinoam et al., 2015; Bucher et al., 2018; Heuser, 1980).

We found that around half of the clathrin coat area has preassembled when plasma membrane invagination begins (Fig. 2 D and Fig. 4 C), agreeing well with previous reports (Bucher et al., 2018; Scott et al., 2018), after which the coat keeps growing gradually. Based on this observation, we developed a mathematical model that for the first time considers that curvature is generated in a positive (nonlinear) feedback loop. Our cooperative curvature model assumes that net area growth of the clathrin coat occurs via triskelia addition to the many available binding sites at the coat edge. We proposed that this process depends on the number of free binding sites, which scale with the edge length and can be described by a single kinetic constant $k_{on}$. Triskelia addition within the coat is still likely to occur at a lower rate, as clathrin lattices can be unsaturated and have defects (Frey et al., 2020; Sochacki et al., 2021), and triskelia can exchange (Avinoam et al., 2015), which is energetically favorable for curvature changes within these lattices (Frey and Schwarz, 2020; Sochacki et al., 2021; Tagiltsev et al., 2021). While growth at the edge accounts for an increase in surface area, the curvature is most likely generated in the bulk of the coat. Here, we assumed a nonlinear relation between the rate of curvature increase and curvature, which reflects cooperativity in the lattice, e.g., due to rearrangements of neighboring triskelia or larger regions thereof. This assumption of cooperativity is supported by recent experiments, which suggest that clathrin exhibits curvature-sensing properties, preferentially assembling

at prebent membranes (Sochacki et al., 2021; Zeno et al., 2021). The model described above shows an excellent fit between theory and experiment and further predicts a square root dependence of θ over time. This describes curvature generation during clathrin coat maturation as a nonlinear mechanism driven by a positive feedback of multiple triskelia, slowing down once the coat approaches a preferred degree of curvature.

Even though our model agrees very well with the data, it still exhibits certain limitations worth mentioning:

First, it only considers the coat itself and ignores the plethora of other proteins within the endocytic machinery. These include numerous BAR domain proteins that are dynamically recruited and disassembled during endocytosis and thus are bound to influence membrane curvature at endocytic sites, as well as a variety of clathrin adaptor proteins, whose presence or absence could explain the cell-type specific differences in average vesicle sizes that we observed (Fig. S4). Taken together, these factors could explain the imperfection of our model in the very beginning and the final part of the timeline (Fig. 4 A), where vesicle scission is driven by the fast-acting mechanoenzyme dynamin. Additionally, we only modeled clathrin recruitment and ignored clathrin disassembly, which could be mediated by adaptor unbinding (Taylor et al., 2011) and uncoating factors, including auxilin (He et al., 2020), that are recruited before the end of vesicle scission. We also assumed that the clathrin coat has constant properties, most notably that the intrinsic coat-driven curvature generation toward its preferred curvature occurs unidirectionally and remains the same throughout endocytosis (Eq. 3). It is however likely that the properties of the clathrin coat change during endocytosis, e.g., by coat stiffening or increasingly tight interactions between triskelia (Frey and Schwarz, 2020; Frey et al., 2020; Sochacki et al., 2021).

Second, we reconstructed an average trajectory of clathrin coat remodeling generated from many individual static snapshots, thereby averaging conceivable different pathways of CME. However, different pathways with substantially changed relationships between the parameters like curvature and θ would be visible in the corresponding plots as separate point clouds. This is not the case, rather we observed a continuous correlation between curvature and θ following a single trajectory, indicating that CME follows a single, stereotypic pathway. We did identify a small, disconnected population of sites in our data set that most likely originate from a distinct cellular mechanism (Fig. S3). While this indicates that our approach should capture potentially different pathways of clathrin-coated vesicle formation, we cannot exclude minor mechanistic variations that are included in the average trajectory.

Clathrin recruitment has been quantified extensively using live-cell microscopy, resolving the sequential recruitment and dynamic characteristics of many important endocytic components (Aguet et al., 2013; Cocucci et al., 2012; Doyon et al., 2011; He et al., 2017; Jin et al., 2021 *Preprint*; Loerke et al., 2009; Mettlen et al., 2009; Saffarian et al., 2009; Schöneberg et al., 2018; Taylor et al., 2011). We wondered if it is possible to correlate our pseudotime reconstruction with previously reported real-time dynamics of clathrin, which shows an initial fluorescence intensity increase, a plateau, and finally a sharp intensity decrease after vesicle scission. While we observed a correlation

between the number of clathrin localizations and surface area (Fig. S5, M–O), we note that indirect immunolabeling is not always quantitative, which complicates a direct comparison of previously reported live-cell fluorescence intensity over time and our pseudotime trajectory data. Nevertheless, we speculate that the initial fast intensity increase in live-cell studies likely corresponds to the initial growth of the flat clathrin coat that escapes our detection due to the fast dynamics and small size. The slower fluorescence increase and subsequent plateauing then coincide with curvature generation and the final addition of triskelia at the coat edge, as resolved in detail in our pseudotime data. To better understand the correlation between changes in the nanoscale architecture of clathrin coats and dynamic consequences, we would ultimately require a method that combines high structural with temporal resolution. In a recent publication, this was attempted using a live-cell 2D super-resolution microscopy approach (Willy et al., 2021). This study reported an increasing projected area of clathrin over time, suggesting that curvature is present in the earliest detectable clathrin structures, and concluded that CME follows the CCM. Although we also find that completely flat structures are rare and curvature is initiated before the final surface is acquired, our data is entirely incompatible with the CCM. This is especially true in the first half of endocytosis with shallow but progressively increasing curvatures (Fig. 3), which is challenging to measure using 2D imaging. This shows that it remains highly desirable to ultimately image the 3D nanoscale architecture of the clathrin coat in living cells in real-time.

In summary, we characterized the dynamics and geometries of the clathrin coat in endocytosis by combining 3D super-resolution microscopy, quantitative analysis, and mathematical modeling. We found that clathrin bending and assembly occur simultaneously along a precisely defined trajectory. We anticipate that this work will be foundational to further study the structural mechanism of endocytosis, both under physiological conditions, and in diseases, where changes in CME likely occur very frequently and have just recently been shown to have profound functional consequences (Moulay et al., 2020; Xiao et al., 2018).

## Materials and methods

### Cell culture

SK-MEL-2 cells (gift from David Drubin, UC Berkeley, described in Doyon et al. [2011]) were cultured adherently in DMEM (Gibco, no. 10565-018), supplemented with 10% (v/v) FBS (Gibco, no. 10270-106), and ZellShield (Biochrom AG) at 37°C, under an atmosphere with 5% $CO_2$ and 100% humidity.

U2OS cells (#300174; Cell Line Services) were cultured adherently as described previously (Thevathasan et al., 2019) in DMEM (Gibco, no. 11880-028) supplemented with 10% FBS (Gibco, no. 10270-106), 1x GlutaMAX (Gibco, no. 35050-038), nonessential amino acids (Gibco, no. 11140-035), and ZellShield (Biochrom AG) at 37°C under an atmosphere with 5% $CO_2$ and 100% humidity.

3T3 mouse fibroblasts (gift from Alba Diz-Muñoz, EMBL Heidelberg) were cultured adherently in DMEM (4.5 g/l D-Glucose) supplemented with 1× MEM NEAA (Gibco, no. 11140-035), 1× GlutaMAX (Gibco, no. 35050-038), and 10%

(v/v) FBS (Gibco, no. 10270-106) at 37°C under an atmosphere with 5% $CO_2$, 100% humidity.

## Sample preparation for superresolution imaging of clathrin-coated pits

Cells were seeded onto high-precision 24-mm round glass coverslips (No. 1.5H, catalog no. 117640; Marienfeld). Coverslips were previously cleaned by incubating them overnight in a methanol/hydrochlorid acid (50:50) solution while stirring. The coverslips were then repeatedly rinsed with water until a neutral pH was reached. They were then placed into a laminar flow cell culture hood overnight to dry. In a final cleaning step, the coverslips are irradiated with ultraviolet light for 30 min.

Cells were fixed as described previously (Li et al., 2018) using 3% (w/v) formaldehyde, 10 mM MES pH 6.1, 150 mM NaCl, 5 mM EGTA, 5 mM glucose, and 5 mM $MgCl_2$ for 20 min. Fixation was quenched in 0.1% (w/v) $NaBH_4$ for 7 min. The sample was washed three times with PBS and permeabilized for 15 min with 0.01% (w/v) digitonin (Sigma-Aldrich) in PBS. The sample was then washed twice with PBS and blocked for 1 h with 2% (w/v) BSA/PBS, washed with PBS, and stained for 3–12 h with anti-clathrin light chain (sc-28276; Santa Cruz Biotechnology) and anti-clathrin heavy chain rabbit polyclonal antibodies (ab21679; Abcam) in 1% (w/v) BSA/PBS. After three washes with PBS, the sample was incubated for 3–4 h with a secondary donkey anti-rabbit antibody (711-005-152; Jackson ImmunoResearch) that was conjugated to Alexa Fluor 647–NHS at an average degree of labeling of 1.5. The sample was then washed three times and mounted for imaging in blinking buffer (50 mM Tris/HCl pH 8, 10 mM NaCl, 10% (w/v) D-glucose, 500 μg ml⁻¹ glucose oxidase, 40 μg ml⁻¹ glucose catalase, and 35 mM MEA in $H_2O$).

For the analysis of the disconnected population of sites, 3T3 cells were transfected with a plasmid encoding the sigma 2 subunit, fused to GFP (gift from Steeve Boulant, University of Florida, #53610; Addgene), to obtain cells transiently expressing AP2-GFP. The transfection was performed using a Lipofectamine 2,000 reagent (Life Technologies) according to the manufacturer's recommendations: 1 μg DNA was mixed with 50 μl OptiMEM I (Thermo Fisher Scientific). The same was done for 3 μl Lipofectamin in 50 μl OptiMEM I. Both solutions were incubated for 5 min and then mixed together and incubated for an additional 10 min at room temperature. The media of previously seeded cells was exchanged to prewarmed OptiMEM I, to which the DNA-Lipofectamin solution (100 μl) was added dropwise. After ~24 h of incubation (at 5% $CO_2$, 37°C), the medium was exchanged with fresh growth medium. After additional incubation for ~16 h, cells were fixed according to the same protocol described above.

## Superresolution microscopy

SMLM images were acquired at room temperature (24°C) on a custom-built microscope (Mund et al., 2018) with a 160× NA 1.43 oil-immersion objective (Leica). The sample was illuminated using a LightHub laser combiner (Omicron-Laserage Laserprodukte) with Luxx 405, 488, and 638 and Cobolt 561 lasers, which were triggered using a Mojo FPGA (Embedded Micro) for microsecond pulsing control of lasers. The lasers were guided through a speckle reducer (LSR-3005-17S-VIS; Optotune,

Dietikon, Switzerland) and coupled into a multimode fiber (M105L02S-A; Thorlabs). The output of the fiber was magnified and imaged in the sample. Fiber-generated fluorescence was removed using a laser clean-up filter (390/482/563/640 HC Quad; AHF). The focus was stabilized using a closed-loop system based on reflecting the signal of a near-infrared laser on the coverslip and detecting the resulting signal on a quadrant photodiode, resulting in focus stabilities of ± 10 nm over several hours. Fluorescence emission was filtered using a 676/37 or a 700/100 bandpass filter (AHF) and recorded by an EMCCD camera (Evolve512D; Photometrics). Typically, 100,000–300,000 frames were acquired using 15- or 30-ms exposure times and laser power densities of ~15 kW/cm². 405-nm laser intensity was adjusted automatically by changing pulse duration to keep the number of localizations per frame constant during the acquisition.

For the analysis of the disconnected population of sites, one diffraction-limited image was additionally acquired before SMLM imaging. For this, a 488 laser at 1.4 kW/cm² was used to take a single frame at 30 ms exposure time. Emission was filtered using a 525/50 bandpass filter (AHF).

The microscope hardware and data acquisition was handled via Micro-Manager 1.4.22 using custom-written software (Edelstein et al., 2010, 2014; Deschamps and Ries, 2020).

## Data analysis

All data analysis was conducted in SMAP ([Ries, 2020] based on MATLAB and available as open source under https://github.com/jries/SMAP).

### *Superresolution image reconstruction*

For fitting the localizations, peaks were detected as maxima in raw camera images after applying a difference-of-Gaussian filter. At those positions, cropped images of 13 × 13 pixels were fitted with an experimentally derived PSF model (free fitting parameters: $x$, $y$, $z$, photons per localization, background per pixel) using an MLE (Maximum likelihood estimation) fitter (Li et al., 2018). The $x$, $y$, and $z$ positions were corrected for residual drift by a custom algorithm based on redundant cross-correlation. In short, the data were distributed into 10-time bins of equal length. For each bin, a superresolution image was reconstructed. We then calculated the image cross-correlations among all superresolution images and extracted the relative displacements in x and y from the position of the maximum in the cross-correlation images. We then calculated the drift trajectory that best describes the relative displacements. In a second step, the z-drift was measured in an analogous way using intensity profiles in z instead of images. Localizations persistent over consecutive frames (detected within 35 nm from one another and with a maximum gap of one dark frame) were merged into one localization by calculating the weighted average of $x$, $y$, and $z$ positions and the sums of photons per localization and background. Localizations were filtered by the localization precision in x,y (0–20 nm) and z (0–30 nm) to exclude dim localizations. The modal value for the localization precision $σ$ was 3.9 nm in x/y and 12.5 nm in z, leading to a resolution estimate (calculated using the Full Width Half Maximum (FWHM) using FWHM = $2\sqrt{(2ln2)}σ$) of 9.2 nm in x/y and 29.4 nm in z (typical

values based on the representative image). Superresolution images were constructed with every localization rendered as a two-dimensional spherical Gaussian with a sigma of 3 nm. The red hot color map used represents the density of localizations and is scaled in a way that 0.03% of the pixel values are saturated.

## Quantitative geometric analysis of clathrin-coated structures

Clathrin-coated structures were segmented semiautomatically. First, we manually defined a region of interest excluding the edges of the cell. Then, the image was blurred using a Gaussian filter with a sigma of 100 nm and peaks were detected using a manually set threshold. This typically yielded several hundreds of sites in a region of 30 × 30 μm. These candidate sites were curated manually, and only single, well-isolated clathrin-coated structures were retained in the dataset.

Next, these structures were analyzed using LocMoFit, an MLE-based model fitting framework that we developed recently (Wu et al., 2023). LocMoFit directly fits localization coordinates with the probability density function (PDF) of a parametrized geometric model. In this study, we modeled clathrin-coated structures with a hollow spherical cap parameterized by the surface area $A$ and the closing angle $\theta$. $\theta$ is defined as the angle between the two vectors that point to the pole and the edge, respectively, from the center of the sphere. The position of the model is defined as the center of mass of the cap. In practice, we discretized the cap by placing spiral points (Saff and Kuijlaars, 1997) or the spherical Fibonacci lattice on the surface of the cap to approximate an even distribution. In LocMoFit, these points were treated as discrete fluorophore coordinates when constructing the PDF of the model. During fitting, additional parameters including the center position and the orientation of the model were determined with respect to the fluorophore coordinates, and an extra uncertainty and the background weight were applied to the PDF. After fitting, the sphere radius is derived as $r = \sqrt{A/2\pi(1 - \cos\theta)}$, projected area as $A_p = \pi sin^2\theta$, and edge length as $\varepsilon = 2\pi sin\theta$. For some flat sites where the fit produced slightly negative curvature values, curvature $H$ and $\theta$ were set to 0 nm$^{-1}$ and 0,° respectively, to approximate them as completely flat.

After model fitting, a second curation step was performed. With this, we ensured that only well-fitted sites are included in the final data set. Sites were excluded if they were upside down (clearly originating from an upper membrane), double-sites with two sites clearly connected to each other or an adjacent flat structure, and large plaques or structures nondistinguishable from an antibody cluster.

## Pseudo-temporal reconstruction of clathrin remodeling during endocytosis

To sort endocytic sites in pseudotime, they were sorted by the closing angle $\theta$, assigning each site a rank index. Flat sites with a manually assigned $H = 0$ nm$^{-1}$ and $\theta = 0$ were all assigned an index of 0. As pseudo-temporal sorting assumes that all sites are part of the same endocytic trajectory, endocytic sites with curvatures above a cell line—specific threshold ($H > 0.016$ nm$^{-1}$ for SK-MEL-2; $H > 0.013$ nm$^{-1}$ for U2OS; $H > 0.014$ nm$^{-1}$ for 3T3) that form a visibly disconnected point cloud (Figs. S3 and S4)

were excluded for this analysis (Fig. S5). To compute pseudo-temporal averages, the sites were spatially and rotationally aligned and rescaled by the average radius of all sites within the respective bin. As the geometric model is rotationally symmetric, we then performed rotational averaging by generating 72 duplicates of each structure, rotating them by 5° with respect to each other, and averaging them. Video 1 was computed using a sliding window approach, where each frame shows an average of 30 sites, and the frame-to-frame increment is 20 sites. The median pseudotime of those 30 sites is indicated.

## Further data analysis

All data that resulted from the quantitative geometric description of clathrin-coated structures were further analyzed in R (R Core Team, 2020). Fitting of the growth models (Tables 1, 2, and 3) was performed according to the equations described in the Appendix using a nonlinear least square fit. All analyses were then performed on filtered data sets, excluding disconnected sites above a cell line—specific threshold ($H > 0.016$ nm$^{-1}$ for SK-MEL-2; $H > 0.013$ nm$^{-1}$ for U2OS; $H > 0.014$ nm$^{-1}$ for 3T3), and sites of negative curvature. During fitting, for sites with $\theta = 0°$, we set $\theta = 0.0001°$ to avoid division by 0.

For depicting the growth models in Figs. 3 and 4, as well as Fig. S5, parameters resulting from the $H(\theta)$ fit were used and mapped to the $A$, $\varepsilon$, and $A_p$ data.

## Simulations

Simulations were performed using the simulation engine for SMLM data implemented in SMAP and LocMoFit as described in Thevathasan et al. (2019). The realistic simulations were based on a two-state (bright and dark) fluorophore model plus bleaching (Sage et al., 2019), and parameters (number of photons, background photons, and fluorophore on-time $t_l$) were extracted from our experiment. (1) First, we defined an equally distributed closing angle $\theta$ from 0 to 180° and calculated the surface area $A(\theta) = 2\pi(1-\cos\theta)/H(\theta)^2$ (see Appendix for details). Here, $H(\theta)$ is defined as in Eq. 3, with fitting parameters determined in SK-MEL-2 (see Table 1). (2) With the defined model parameters, we generated protein positions for each simulated structure by taking randomly drawn $N \propto A$ samples from the PDF of the hollow spherical cap with no uncertainty. (3) With a probability $p_{label} = 0.6$, a fluorescent label was created at a protein position. (4) Linkage displacements in x, y, and z were added to a label and were determined as normally distributed random variables with a variance corresponding to the linkage error of 5 nm. The fluorophore is assumed to be freely rotating three-dimensionally between different blinks. (5) Each fluorophore appeared at a random time and lived for a time $t_l$, determined as a random variable from an exponential distribution with a mean of 1.6 frames. (6) A label had a probability $p_{react} = 0.5$ to be reactivated and then appeared at a random later time point, otherwise it was bleached. (7) When it was on, a fluorophore had a constant brightness. Thus, the brightness in each frame was proportional to the fraction of the time the fluorophore was on in each frame. (8) The emitted photons in each frame were determined as a random Poisson variable with a mean value corresponding to the average brightness in the frame. (9) For

each frame, we calculated the CRLB (Cramér-Rao lower bound) in x, y, and z from the number of photons (with a mean of 11,000) and the background photons (130 per pixel) based on the theoretical Gaussian PSF (Mortensen et al., 2010) or a 3D cspline PSF model derived from beads calibrations (Li et al., 2018). This error was added to the true x, y, and z positions of the fluorophores as normally distributed random values with a variance corresponding to the respective calculated CRLB.

### Online supplemental material

Fig. S1 provides examples of diverse clathrin coat structures. Fig. S2 provides simulations. Fig. S3 shows non-endocytic clathrin structures. Fig. S4 shows clathrin coat remodeling in three different cell lines. Fig. S5 provides linkage error investigation and number of localizations per clathrin coat. Video 1 shows a pseudo-temporal movie of clathrin coat remodeling during endocytosis in SK-MEL-2 cells. Appendix shows a detailed description of the cooperative curvature model. All presented data are available in the BioStudies database (https://www.ebi.ac.uk/biostudies/) under accession number S-BIAD566.

## Acknowledgments

We thank the entire Ries and Kaksonen labs for fruitful discussions and support.

This work was supported by the European Research Council (ERC CoG-724489 to J. Ries), the National Institutes of Health Common Fund 4D Nucleome Program (Grant U01 to J. Ries), the Human Frontier Science Program (RGY0065/2017 to J. Ries), the EMBL Interdisciplinary Postdoc Programme (EIPOD) under Marie Curie Actions COFUND (Grant 229597 to O. Avinoam), the European Molecular Biology Laboratory (M. Mund, A. Tschanz, Y.-L. Wu and J. Ries), and the Swiss National Science Foundation (grant 310030B_182825 and NCCR Chemical Biology to M. Kaksonen). O. Avinoam is an incumbent of the Miriam Berman Presidential Development Chair.

Author contributions: M. Mund, O. Avinoam, M. Kaksonen and J. Ries conceived the study, M. Mund, A. Tschanz, J.L. Mehl and O. Avinoam performed experiments, M. Mund, A. Tschanz, Y.-L. Wu, J.L. Mehl, O. Avinoam and J. Ries analyzed superresolution data; F. Frey and U.S. Schwarz developed the cooperative curvature model; J. Ries supervised the study. M. Mund, A. Tschanz, F. Frey, J. Ries wrote the manuscript with input from all authors.

Disclosure: The authors declare no competing financial interests.

Submitted: 10 June 2022

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

Figure S1. **Examples of diverse clathrin coat structures. (A)** Large clusters of clathrin molecules excluded from further analysis. Shown in top view (xy) with the dotted line indicating a 50 nm-thick z-slice (xz) shown below. **(B)** Vesicular structures are sometimes fitted with a lower θ than expected. **(C)** While a spherical model describes the structure of most endocytic clathrin coats faithfully, there are few cases, as exemplified here, where the elliptical and irregular shape of an assembling coat is difficult to approximate with a simple geometric model. Two orthogonal 50-nm-thick z-slices are shown here in xz and yz, and the respective spherical model fit is plotted as a dotted line. **(D)** Non-continuous labeling of clathrin manifests itself as holes in the coat, indicated with a blue arrow. All scale bars are 100 nm.

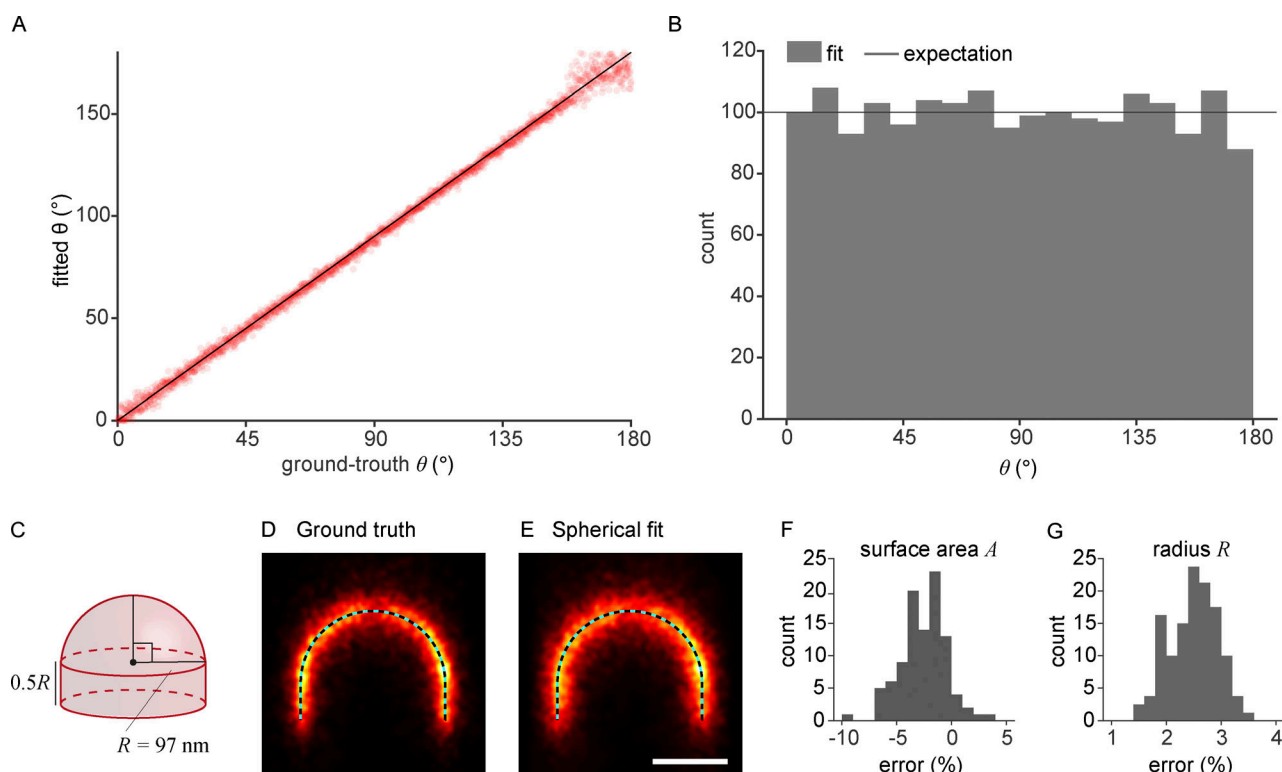

Figure S2. **Simulations. (A and B)** Estimation error of the closing angle $\theta$ ($N$ = 1,800 sites). **(A)** The comparison of fitted $\theta$ of simulated clathrin coat structures to corresponding ground-truth $\theta$ shows no systematic bias and a narrow spread of the error across the ground truth although the spread increases when $\theta <$ 20° or $\theta >$ 160°. We reasoned that in the earlier range, where the structures are flat, the slightly increased error corresponds to the insensitivity of $\theta$ to flat structures. In the later range, where the vesicles are almost closed, the error was caused by indistinguishable tiny holes corresponding to the real vesicle openings and unlabeled clathrin in the coat. The fitted structures were simulated to have similar quality as the experimental data and to distribute evenly across $\theta$. **(B)** The distribution of $\theta$ shows no significant error compared to the expectation corresponding to the evenly distributed $\theta$, except for the small potential underestimation of entirely closed coats. **(C–G)** Averaging preserves U-shapes in simulated endocytic sites. **(C)** A ground-truth structure was used for simulating U-shaped clathrin coats. The model for simulation was built by combining a hemisphere with a radius of R = 97 nm and a cylinder with a height D = 0.5 R. We choose the radius value according to the median radius of bin 6 in Fig. 4 G. This bin has a closing angle slightly larger than 90°. 20-nm-thick cross-sections of the averages ($N$ = 100 sites) registered based on (D) the ground truth and (E) the spherical fit are shown with the ground-truth U-shaped model (dotted line). Histograms of normalized estimation errors are shown for the parameters (F) surface area and (G) radius, with mean values of −2.6 and 2.5% respectively. The scale bar is 100 nm.

Figure S3. **Non-endocytic clathrin structures.** Results of LocMoFit analysis of clathrin structures in SK-MEL-2 cells ($N$ = 13 cells, $n$ = 1,798 sites). **(A)** A strong correlation between curvature and θ can be observed for most structures ($n$ = 1,645 sites, black). A disconnected point cloud ($n$ = 153 sites, 8.5%, red) indicates the presence of endocytosis-unrelated clathrin structures. **(B–D)** The same distinct population of data points can be observed for the (B) surface area, (C) edge length, and (D) projected area. **(B)** Example structures from the disconnected population of sites in top view (xy) and 50 nm-thick z-slices (xz) and their respective fitted θ values. Scale bar, 100 nm. **(C–F)** Analysis of clathrin coats not following general trajectory of curvature generation. **(C)** 3T3 cell transiently overexpressing AP2-GFP. Left: Single-molecule localization microscopy image of immunolabeled clathrin. Middle: Diffraction-limited image of the AP2-GFP signal. Right: Overlay of the two targets (scale bar, 10 µm). **(D)** Enlarged image of the section indicated in C (scale bar, 1 µm). **(E)** Example sites indicated in B (scale bar, 100 nm). 1: Example for a structure annotated as "GFP positive." 2: Example for an "inconclusive" GFP signal. 3 and 4: Example of "GFP negative" structures." **(F)** Analysis results when estimating θ and curvature from clathrin structures and annotating them depending on their AP2 signal ($N$ = 3 cells and $n$ = 277 sites). No AP2-GFP positive structures are found in the disconnected population of sites, suggesting that they are most likely not generated via clathrin-mediated endocytosis.

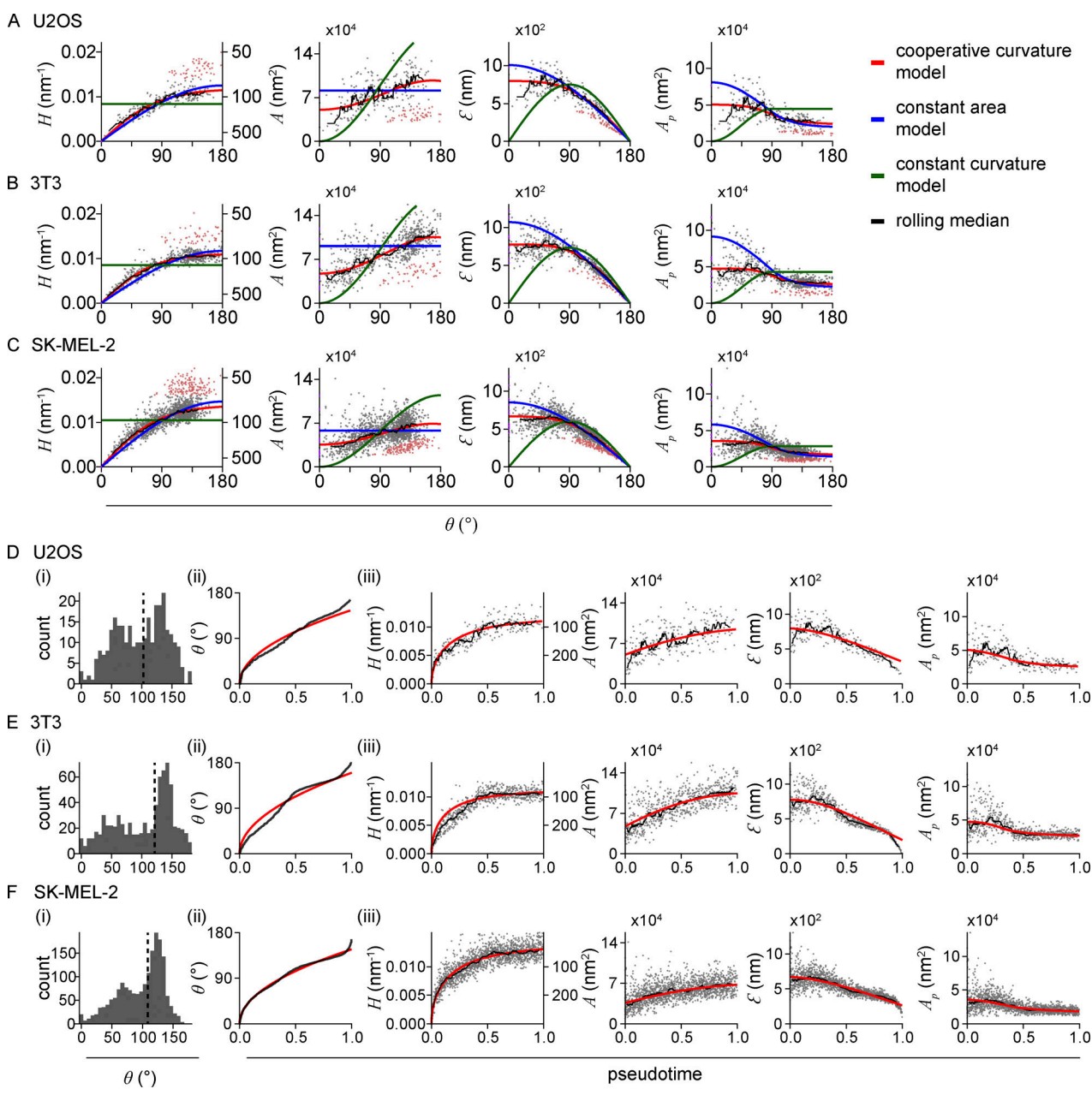

**Figure S4. Clathrin coat remodeling in three different cell lines. (A–C)** Results of LocMoFit analysis for clathrin structures. Different growth models are fitted to curvature H over θ. The resulting fitting parameters are then used to map the same models also over the surface area, edge length, and projected area (left to right). Purple: Completely flat sites with H = 0 nm⁻¹ and θ = 0. Red: Disconnected sites that were excluded from the fitting. Black line: Rolling median (window width = 5% of total number of sites; A) U2OS (*N* = 3 cell, $n_{grey}$ = 241 sites, $n_{red}$ = 53 disconnected sites, $n_{purple}$ = 1 completely flat sites), (B) 3T3 mouse fibroblasts (*N* = 7 cells, $n_{grey}$ = 688 sites, $n_{red}$ = 51 disconnected sites, $n_{purple}$ = 8 completely flat sites), and (C) SK-MEL-2 cells (*N* = 13 cells, $n_{grey}$ = 1,631 sites, $n_{red}$ = 153 disconnected sites, $n_{purple}$ = 14 completely flat sites). **(D–F)** Temporal reconstruction of clathrin coat remodeling. (i) Distribution of θ slightly differs between cell lines, especially in the earlier states. Median θ shown as dotted lines correspond to 99.6 for U2OS; 121.4° for 3T3; and 108.5° for SK-MEL-2 cells. (ii) The cooperative curvature model (red line) highlights the square-root dependence between θ and pseudotime. (iii) The cooperative curvature model is used to describe the curvature H propagation over pseudotime. Resulting fitting parameters are then used to map the same model to surface area A, edge length Ɛ, and projected area $A_p$. A rolling median is plotted in black (window width = 5% of total number of sites).

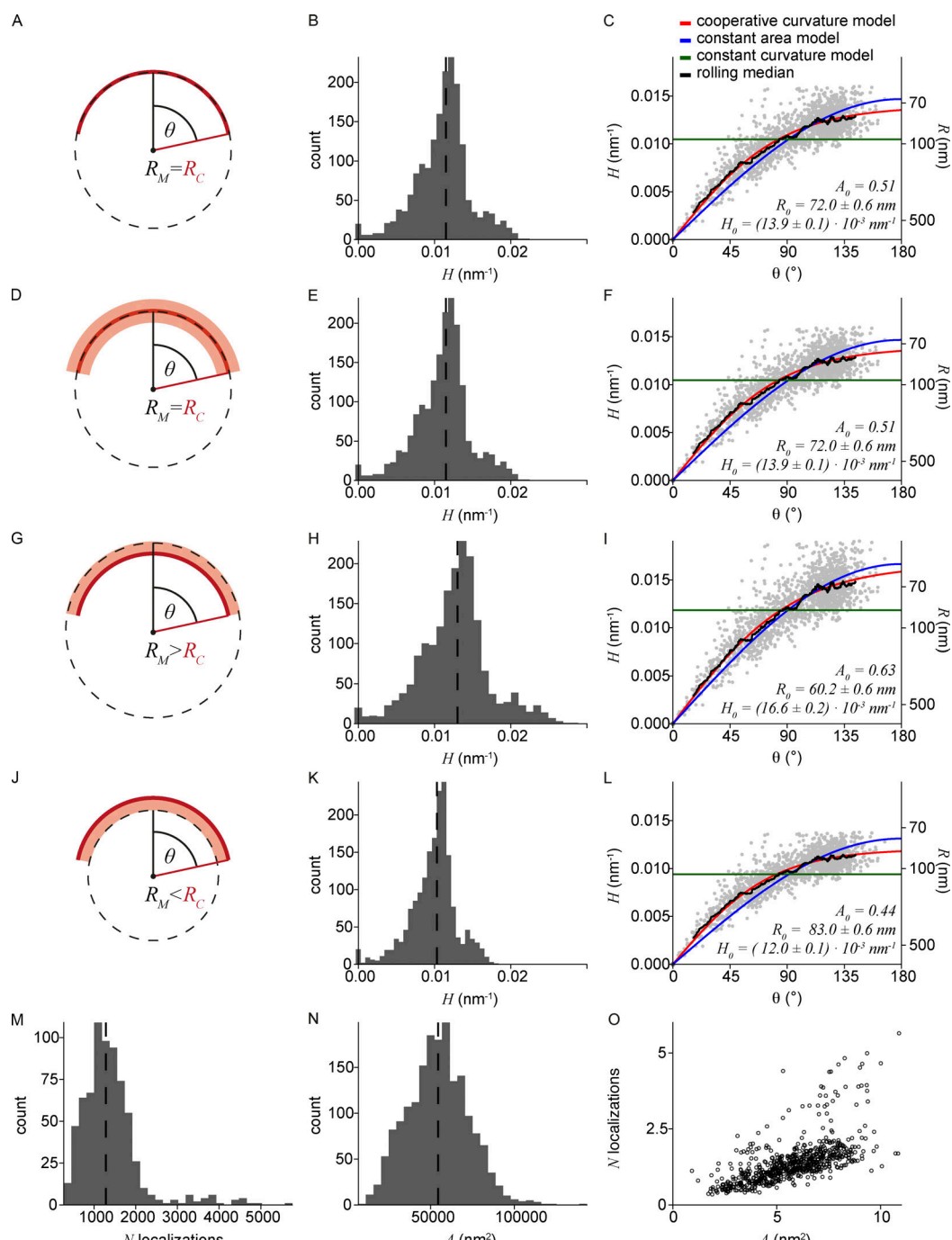

Figure S5. **Linkage error investigation and number of localizations per clathrin coat. (A–L)** Impact of linkage error on geometric model fit. Indirect immunolabeling displaces the label from the target molecule by two antibodies. This generates a so-called linkage error of on average ± 10 nm, and resulting localizations might not accurately represent the underlying structure of interest (Früh et al., 2021). **(A)** An ideal case of no linkage error, where the model (dotted black line) with radius $R_M$ accurately represents the underlying clathrin coat (red) with a radius $R_C$. **(B)** Histogram of quantified curvature with a median of 0.011 nm⁻¹. **(C)** Models fitted to curvature propagation over θ. **(D)** Uniform displacement of localizations (light red, ± 10 nm) due to unbiased labeling by the antibodies. The radius $R_M$ still accurately represents the true radius $R_C$. **(E)** Histogram of the curvature, assuming no needed correction of $R_M$. Median: 0.011 nm⁻¹. **(F)** Models fitted to curvature propagation over θ remain the same as for C. **(G)** Biased labeling of the antibodies (light red, + 10 nm) could result in an overestimation of $R_M$ by 10 nm. **(H)** Histogram of the curvature corrected by subtracting 10 nm from quantified $R_M$. Median: 0.013 nm⁻¹. **(I)** Models fitted to corrected curvature propagation over θ. **(J)** Biased labeling could result in an underestimation of $R_M$ by 10 nm. **(K)** Histogram of the curvature corrected for the overestimation in radius by adding 10 nm from quantified $R_M$. Median: 0.010 nm⁻¹. **(L)** Models fitted to corrected curvature propagation over θ. **(C, F, I, and L)** Fitting parameters are $A_0$: Fraction of surface area growing as a flat lattice before curvature initiation, defined as $A_0 = A(θ = 0.01)/A(θ = π)$; $R_0$: Preferred radius of the clathrin coat fitted with CoopCM; $H_0$: Preferred curvature of the clathrin coat fitted with CoopCM. Analysis of $N$ = 13 SK-MEL-2 cells, $n$ = 1,645 sites. While the fitting parameters scale with the error in radius estimation, the relationships among the parameters and thus our mechanistic interpretation by the cooperative curvature model still holds true. **(M–O)** Number of localizations versus surface area. For $N$ = 6 SK-MEL-2 cells, $n$ = 700 sites the number of localizations found in one clathrin-coated structure was extracted. This is plotted against the quantified surface area determined for each coat.

Video 1.   **Pseudo-temporal movie of clathrin coat remodeling during endocytosis in SK-MEL-2 cells.** Each frame corresponds to a sliding window average of 30 sites, with a frame-to-frame increment of 20 sites. The median pseudotime of those 30 sites is indicated.

**Provided online is an Appendix, containing a detailed description of the cooperative curvature model. Further, all data shown here are also available in the BioStudies database (https://www.ebi.ac.uk/biostudies/) under accession number S-BIAD566.**

