## [Peer Review File · The Journal of Cell Biology]

Clathrin coats partially preassemble and subsequently bend during endocytosis

Markus Mund, Aline Tschanz, Yu-Le Wu, Felix Frey, Johanna Mehl, Marko Kaksonen, Ori Avinoam, Ulrich Schwarz, and Jonas Ries

Corresponding Author(s): Jonas Ries, European Molecular Biology Laboratory

Review Timeline:	Submission Date:	2022-06-10
	Editorial Decision:	2022-07-06
	Revision Received:	2022-11-29

Monitoring Editor: Pier Paolo Di Fiore

Scientific Editor: Tim Spencer

Transaction Report:

DOI: <https://doi.org/10.1083/jcb.202206038>

Revision 0

Review #1

1. Evidence, reproducibility and clarity:

Evidence, reproducibility and clarity (Required)

This is a well-executed and interesting study addressing a still controversial issue in clathrin-mediated endocytosis, namely the nature of curvature generation during formation of endocytic clathrin coated vesicles. The authors have applied new techniques to this old question, including state-of-the-art high resolution 3D single-molecule localization microscopy (SMLM, i.e. Super-resolution microscopy), a new maximum-likelihood based fitting framework to fit complex geometric models into localized point clouds (Wu et al., 2020, BioRxiv) and mathematical modeling leading to a new cooperative curvature model of clathrin coat remodeling and temporal reconstruction of CCP structural dynamics based on the distribution of static super-resolution images. This is an important contribution, but will it resolve the controversy of constant curvature vs constant area for CCP invagination? I doubt it. In some ways the controversy is somewhat contrived and, as this paper shows the answer is unlikely to be either or. Below are some specific comments, in somewhat random order, from someone (a curmudgeon?) who has reviewed and/or carefully read these papers since 1980. Points that the authors should address are in bold. All can be addressed with modifications to the text, as the one experiment I asked for (quantification of clathrin recruitment) is impossible with this approach).

1. I wonder how many people who cite Heuser's 1980 paper have ever read it carefully. Indeed, many of the observations made here were also made by Heuser. Below, for example, is a summary I wrote, but then removed from a review as it was too lengthy

"While Heuser favored the model that CCPs assemble first as flat structures and then rearrange during invagination, he was also careful to note several caveats. First, he observed that the edges of CCPs were 'ragged', likely reflecting sites of assembly of new polygons and that pentagons were more abundant at the edges. Thus, he argued that 'if even a few of these edge pentagons were destined to become completely surrounded with hexagons, it would be necessary to conclude that some degree of curvature can be built into coats as soon as they form'. Second, by examining tilted sections he observed that "even the flattest baskets have a small degree of inward curvature, and many were complete hemispheres". Finally, he cautioned that his images were snap-shots and a precursor-product relationship could not, therefore, be unambiguously established and that the very large flat lattices he observed might well be 'prove to be some sort of dead end'. We now know that fibroblasts, in particular, have large numbers of static flat clathrin plaques."

Thus, many of the author's conclusions, i.e. that 'completely flat clathrin coats are rare (pg 12, although they're not numbered), and that curved structures can be seen to emerge from the edges

of flat lattices (see Supplemental Figure 1a, 3 examples on the right) are indeed consistent with Heuser's observations. In many ways, Heuser's 1980 paper is used as a straw man argument for the constant area model. The authors should more accurately cite and acknowledge this seminal paper.

2. As Heuser did in his 1980 classic, the authors here would do well to note several caveats related to their analyses. These include:

- a. Like Heuser they have assembled static imaged to create a pseudotemporal model, albeit using a much more quantitative approach. Nonetheless, it seems that this assumes only a single, stereotypic pathway for CCP formation. How good is this assumption? We know from dynamic imaging that there exists significant heterogeneity in both the kinetics and the molecular composition of CCPs. The authors should acknowledge this limitation.

- b. The method, which required that they 'optimized the sample preparation to densely label clathrin at endocytic sites' involves labeling cells to near saturation with rabbit polyclonal antibodies to both clathrin light chains and clathrin heavy chains followed by detection with a second polyclonal donkey anti-rabbit. This gives 20 nm of additional and presumably flexible linker on the label. How might this effect the measurements and modeling? The Wu et al paper, which BTW has not been peer-reviewed, shows high precision fitting of the nuclear pore structure, but using endogenously tagged NUP-95, not two-layers of antibodies. The authors will need to discuss this limitation, it is my biggest concern regarding the analysis shown.

3. One reason for continued controversy in this field is the lack of any attempt to resolve findings obtained using different methods. Can a parsimonious explanation be found, or are their artifacts or misinterpretations of previous findings that can explain the discrepancies? Any valid model should fit all of the valid data. For example, the authors fail to cite a recent paper by Willy et al in *Dev Cell* (PMID 34774130), which has been on BioRxiv since 2019 (doi: <https://doi.org/10.1101/715219>). Here, similar to this present study, the authors used high resolution SIM-TIR to analyze ~1000 CCPs in 3 different cell lines (sadly non-overlapping with the cells used herein) and in *Drosophila* embryos to quantitatively test the two models. They conclude that their findings unambiguously support a constant curvature model. The authors would do the field a favor if they carefully read this paper and identified areas of commonality (i.e. that curvature is detected at early stages in both cases) and possible explanations for the discrepancies. Certainly, they should not ignore it.

4. An important body of evidence that is not considered in their model or discussion is that derived from live cell imaging. In addition to the heterogeneity mentioned above, studies have shown that the clathrin addition to CCPs is complete (i.e. the growth phase) occurs within the first ~20-30s, followed by a variable length (0->100s) plateau phase (Loerke et al, PMID 21447041). Both the current study and the Willy et al study admit that they may not be able to detect the earliest intermediates in CCP assembly. Indeed, in this study the smallest surface area CCPs are only 2-fold smaller than the largest CCPs, suggesting that over half of the triskelions have been recruited before a CCP can be distinguished from the background of clustered, nonspecifically-bound antibodies. Could the authors be monitoring events during the plateau phase and not the earliest events? Regardless, the findings are important as they address the nature of curvature generation during this plateau phase. While monitoring curvature generation during early events in CME, a recent study (Wang et al., *eLife*, PMID 32352376) showed that the acquisition of curvature within the first 20s of CCP assembly was a distinguishing feature between abortive and productive events. The authors might discuss how these studies on CCP dynamics might (or might not) inform their models.

5. The authors advertise 'quantitative' description of clathrin coated structure and indeed their measurements and models are quantitative; but there is no measure of intensity/numbers of triskelions and CCP growth: an important piece of quantitative data. I expect this is impossible with indirect immunofluorescence but should be considered as a limitation of the approach. Indeed, to my knowledge no one has yet quantitatively measured curvature generation in parallel to clathrin addition at CCPs (closest is Saffarian and Kirchhausen, PMID 17993495), but they don't discuss the relationship.
6. On page 7 equation 1, you assume a constant growth rate for addition of triskelia, but later describe that the rate might be cooperative (as the number of edges increases). How would this affect your modeling?

****Minor points:****

- Can you indicate in the first paragraph of the results that you are using indirect immunofluorescence with rabbit anti-CLCA, anti-CHC and detection with donkey anti-rabbit for labeling, to augment the rather vague statement 'we optimized the sample preparation to densely label clathrin at endocytic sites'.
- I'm not comfortable with the conclusion on page 5 that your data 'indicates that at the time point of scission, the clathrin coat of nascent vesicles is still incomplete'. Other explanations might be the relative kinetics of scission vs CCP growth (i.e. these structures are too transient to detect), or that deeply invaginated pits are sheered-off the membrane during sample preparation (there is evidence that most biochemically isolated CCVs are derived from sheered CCPs).
- Bottom of page 5, can you briefly mention what data is shown in Supplemental Figure 2 (ie. Figure 2D and examples of likely non-endocytic CCPs shown in Supplemental Figure 2). When I read this, I questioned your speculation.
- Can you indicate N CCPs from N cells in the data in Tables 2-3 for fibroblasts and U2OS cells? Do you observe and have to ignore a larger number of flat/clustered CCPs in the fibroblasts?
- The last 3 paragraphs of the Introduction are results. The Introduction might best be used to review literature in more detail, discuss the reasons why uncertainty still exists and perhaps indicate how the methods applied here will help.

2. Significance:

Significance (Required)

This is another excellent addition to a growing list of papers seeking to define the process of curvature generation at endocytic clathrin coated pits. In my opinion, its impact would be increased by better integrating the results presented here with other studies and methods, including the recent paper by Willy et al and the large body of literature on coated pit dynamics, some of which might be relevant in interpreting results, or at least placing them in a real vs pseudo-temporal perspective. The methods introduced and the quality of imaging, modeling and quantification further increase the study's significance. The finds will be of interest to those in the CME field, those studying membrane curvature generation in other contexts, those modeling CME, vesicle formation and curvature generation and those using SMLM to discern the structure of macromolecular assemblies.

Reviewer expertise: Clathrin-mediated endocytosis (Sandra Schmid)

3. How much time do you estimate the authors will need to complete the suggested revisions:

Estimated time to Complete Revisions (Required)

(Decision Recommendation)

Less than 1 month

Review #2

1. Evidence, reproducibility and clarity:

Evidence, reproducibility and clarity (Required)

****Summary****

In this article, the authors aimed to investigate the dynamic of clathrin lattice during clathrin-mediated endocytosis (CME). Overall, they successfully achieved the goal by observing a large number of clathrin spots from several cell lines with 3D single-molecule localization microscopy (SMLM). With the help of this high-resolution imaging technique, they were able to describe the physical properties of each spot and reconstruct the assembly and remodeling of the clathrin coat. Moreover, by comparing the constant area/curvature model with their own data, the authors highlighted that neither of the prevailing models perfectly explained what they observed and proposed 'cooperative curvature model'. With the novel model, the authors were able to reconstruct the clathrin coat remodeling in different cell lines and concluded that the simultaneously bending and assembly of the clathrin coat is a homogenous property of endocytosis.

The experiments and analytical procedures are well-designed and performed, and the manuscript is well-organized. The conclusion 'cooperative curvature model' was deduced from a large amount of data analysis and clearly stated in the text. I would like to recommend its publication if the following issues will be clarified.

****Major comments:****

1. The authors compared the morphological dynamics of clathrin-coated pit among three different cell lines (SK-MEL-2, U2OS, and 3T3) and found slight differences. As U2OS cells was derived from bone tissues, it has different mechanical properties (membrane tension, elasticity of cortical layer, etc..). It would be interesting to consider those mechanical properties

in understanding the morphology (Figure 2) and progress (Figure 4) of the CME. Considering the fact that the bending energy of the plasma membrane is dependent on the membrane tension, they may be able to find some relationships between mechanical properties of the cell cortex and CME.

2. In Figure 4, the authors estimated the progression of the CME using the frequency distribution of theta. However, I wonder how they handled the events which were aborted in the middle of the CME. It had been suggested that some CME are aborted during the initial step of the CME. The authors should consider (at least discuss) those abortive events, which can disturb the analysis.

****Minor comments:****

1. Page5, result section 2. The author should further explain why vesicles from trans Golgi could be responsible for the small disconnected set of data points corresponding to the vesicles with larger curvatures.

2. Page7, line 6. The author assumed that the clathrin coat starts growing on a flat membrane. However, as is mentioned in the discussion, clathrin has been proved to have curvature sensing ability which could be further amplified by adapter proteins by several times (Zeno et al., 2021). So, it seems that clathrin preferred a highly curved membrane instead of a flat one. Is it still reasonable to make this assumption?

3. Page 9, result section 4. In the sentence: "we effectively generated the average trajectories of how curvature, surface area, projected area and lattice edge change during endocytosis in SK-MEL-2 cells (Figure 4B-E)." Here I think the authors are describing Figure 4C-F.

4. Page 11, discussion. In the sentence: "A deviation of the cross-sectional profile from a circle is nevertheless preserved in the averaging (Supplementary Figure 5)." I didn't see supplementary figure 5 in the article.

2. Significance:

Significance (Required)

From a vast amount of microscopic images and data analysis, the manuscript gives a clear model on the progress of the CME, which integrates two opposing models; constant area and constant curvature models. This is a big progress in our understanding of the molecular mechanism of CME, and will attract many researchers in the field of cell biology. From a viewpoint of my expertise (molecular imaging of plasma membrane and endocytic processes), this manuscript has significant impact on the related research fields.

3. How much time do you estimate the authors will need to complete the suggested revisions:

Estimated time to Complete Revisions (Required)

(Decision Recommendation)

Between 1 and 3 months

Review #3

1. Evidence, reproducibility and clarity:

Evidence, reproducibility and clarity (Required)

****Summary:****

The authors used single-molecule localization microscopy of clathrin in fixed cells (2 human cell lines, one mouse) to capture snapshots of a clathrin-mediated endocytosis (CME), fitted these localizations to a geometric model of a forming vesicle, and used these fitted measurements to test existing models of clathrin-mediated vesicle formation before refining their own.

Specifically, the closing angle, a measure of vesicle completeness, was used as a proxy for growth-stage of the vesicle such that the many captured snapshots could reconstruct a pseudo-timeline with an unknown parameterization of time on closing angle. Two standard models of CME vesicle formation, where the surface area is kept constant or where the curvature is kept constant, were examined and determined to be incommensurate with the pseudo timelines of curvatures and surface area. The authors then describe their own model for CME vesicle formation, in which neither surface area nor curvature are constant in evolution of the vesicle, and cooperative forces are hypothesized to non-linearly modulate the curvature-growth as a function of closing angle. Additionally, by binning snapshots and then aligning, scaling, and azimuthally smoothing each bin, they reconstruct representations of distinct endocytic stages.

****Major comments:****

Most results are quite convincing, and the authors do a nice job of displaying examples of SMLM data, both with fit results as well as example clathrin assemblies that are too far removed from their budding-vesicle model to be included for analysis, for example. It is also worth noting that the clathrin images themselves appear to be very high-quality - clearly, as detailed in the methods, attention was given to each step of the imaging and reconstruction process.

While the presented cooperative curvature model seems reasonable and surely fits the curvature-, surface area-, and rim length-vs. closing angle data better than the simplistic constant surface-area and constant curvature models, it also has more parameters, namely: γ (the initial rate of curvature change with closing angle) and H_0 (the final preferred curvature). It would be appropriate to calculate an information criterion (e.g. Bayesian), using an assumption of Gaussian-distributed errors (presumably the data fitting in R was least squares, so this would match) to justify the additional parameters.

A related issue relates to the error in the extracted value of the closing angle from a single 3D reconstruction - the error distribution should be quantified for this very important parameter. The

errors in the other parameters extracted from the fits are less important, but would enhance the paper.

Pseudo-temporal sorting on closing angle makes sense and I appreciate the authors mentioning potential caveats to the monotonicity, etc. However, a comment about the impact of closing angle errors on the pseudo-time determinations would be helpful. The agreement of theta-rank plots with the hypothesized \sqrt{t} scaling is reassuring.

I additionally appreciate the robustness of fitting a geometric structure from localizations rather than relying on pseudo-temporal sorting on clathrin count extracted from localization-merging of multi-blinking emitters.

The authors did a nice job of qualifying their more speculative claims, in particular I appreciated their mentioning the possibility that smaller clathrin coats could be below their detection limit.

The authors state a set of data points in suppl. figure 2D (and suppl. Fig 3A-C) are "likely" small clathrin-coated vesicles from the trans Golgi. I appreciate the examples rendered in that figure so a reader can appraise, but if they have my background they might not know how reasonable exclusion of this data is from model testing. This claim could be rephrased or the rationale expanded upon to justify the Golgi hypothesis.

The data and methods are presented such that they could be reproduced, and replicating their experiment in multiple cell lines, across multiple species, would seem to be adequate replication. As mentioned above, the statistical analysis of whether the model complexity is justified by improved goodness of fit is currently missing but can readily be checked and added.

****Minor comments:****

Last paragraph of the introduction, positive feedback is mentioned but not the slowing down as preferred curvature is realized (inclusion of which might help foster a clearer understanding of the model early on).

In Fig. 1, please state in the figure caption what is being displayed in the two large panels and what is the color map. Is this the 3D data from the overlapping elliptical Gaussians projected on the plane in a "hot" map? Further, in the top right small panels, are the x-y images projections of all z, or measured at a specific z?

In Eqn. (1), epsilon is not defined.

For the theta-rank plots (Fig4 B, SFig D-F ii) moving the $\theta(t)=\sqrt{t}$ red curves behind sorted theta data would make the data easier to see.

"Laser" in sentence about the speckle reducer should probably be plural.

I would like to see the "custom" algorithm based on redundant cross-correlation for drift correction briefly described.

A legend for supplemental figure 3 A-C would be nice.

I would enjoy hearing the authors' thoughts on why resting points at closing angle 70° and 130° are present. If these thoughts can be readily rationalized/referenced some speculation might even be warranted in the text.

If the definition of the abbreviation flat-to-curved-transition as FTC was explicit I missed it.

Resolution of 20 and 30 nm (laterally and axially, respectively) was quoted once towards the beginning of the manuscript as being an improvement resulting from the localization method described in Li et al., 2018. Resolution can be difficult to speak about precisely, but the methods section would seem to indicate that localizations are filtered at 20 nm lateral localization precision (potentially 30 nm axially?), and I think the authors could consider rephrasing to depict this unless I am missing elsewhere a description of the resolution metric being used.

2. Significance:

Significance (Required)

Proteins involved with inducing curvature in membranes are in general very exciting targets for localization microscopy, yet still for many systems questions remain unanswered. The authors tackle one such question in this manuscript. In other, unresolved, discussions, the posed hypotheses are quite similar to the simplistic models surpassed in this work (e.g. that curvature scales linearly with local protein copy number, or that surface area scales linearly with local protein copy number). The idea of cooperativity may be useful for others to consider, and the authors additionally demonstrate a seemingly smooth workflow using their separately described tools (primarily LoMoFit; Wu et al. 2021).

I myself am not an expert on CME or vesicle trafficking. My background is primarily in SMLM method development and SMLM / fluorescence image analysis. From my perspective, the novelty of the biological conclusions appears to be the authors' specific cooperative model and the presence of two structural states which are enriched (closing angle 70° and 130°). As referenced, and authors F. Frey and U. S. Schwarz nicely present in Bucher et al. 2018, the constant curvature and constant surface area models are known to be inaccurate descriptions of CME evolution, and further it is also known that clathrin first assembles small flat structures before beginning to curve the membrane. However, the 3D super-resolution imaging and direct evaluation of a 3D model geometry in this work is a nice extension of the 2D super-resolution imaging and projection evaluation in the authors' previous work studying endocytosis through ensemble averaging in yeast (Mund et al. 2018) as well as the analysis on projections in Bucher et al. 2018. Fully 3D treatment of the clathrin structures allows the authors to orient asymmetric assemblies such that they are averaged out in their ensemble reconstruction, and as they point out the molecular specificity afforded by a fluorescence-based technique ensures unbiased segmentation of clathrin-involved endocytic sites. In other words, while this work does not describe a technical advance not already described elsewhere, it sets a nice example for those researching protein-membrane interactions of how to leverage the right tools to clearly and directly answer their questions. With their additional work to make these

tools extensible to other geometries, multiple color channels, etc., I expect their work to inspire quality studies in other systems. That significance is complementary to their proposal of a reasonable model for the geometric evolution of CME.

****References:****

Maximum-likelihood model fitting for quantitative analysis of SMLM data
Yu-Le Wu, Philipp Hoess, Aline Tschanz, Ulf Matti, Markus Mund, Jonas Ries
bioRxiv 2021.08.30.456756; doi: <https://doi.org/10.1101/2021.08.30.456756>

Bucher, D., Frey, F., Sochacki, K.A. et al. Clathrin-adaptor ratio and membrane tension regulate the flat-to-curved transition of the clathrin coat during endocytosis. *Nat Commun* 9, 1109 (2018).
<https://doi.org/10.1038/s41467-018-03533-0>

Markus Mund, Johannes Albertus van der Beek, Joran Deschamps, Serge Dmitrieff, Philipp Hoess, Jooske Louise Monster, Andrea Picco, François Nédélec, Marko Kaksonen, Jonas Ries, Systematic Nanoscale Analysis of Endocytosis Links Efficient Vesicle Formation to Patterned Actin Nucleation, *Cell*, 174, 4, (2018). <https://doi.org/10.1016/j.cell.2018.06.032>.

3. How much time do you estimate the authors will need to complete the suggested revisions:

Estimated time to Complete Revisions (Required)

(Decision Recommendation)

Less than 1 month

Full Revision

Manuscript number: RC-xx-xx

Corresponding author(s): First name, Last name

[Please use this template only if the submitted manuscript should be considered by the affiliate journal as a full revision in response to the points raised by the reviewers.]

*If you wish to submit a preliminary revision with a revision plan, please use our "Revision Plan" template. **It is important to use the appropriate template to clearly inform the editors of your intentions.**]*

1. General Statements [optional]

This section is optional. Insert here any general statements you wish to make about the goal of the study or about the reviews.

This section is mandatory. Please insert a point-by-point reply describing the revisions that were already carried out and included in the transferred manuscript.

Reviewer #1 (Evidence, reproducibility and clarity (Required)):

This is a well-executed and interesting study addressing a still controversial issue in clathrin-mediated endocytosis, namely the nature of curvature generation during formation of endocytic clathrin coated vesicles. The authors have applied new techniques to this old question, including state-of-the-art high resolution 3D single-molecule localization microscopy (SMLM, i.e. Super-resolution microscopy), a new maximum-likelihood based fitting framework to fit complex geometric models into localized point clouds (Wu et al., 2020, BioRxiv) and mathematical modeling leading to a new cooperative curvature model of clathrin coat remodeling and temporal reconstruction of CCP structural dynamics based on the distribution of static super-resolution images. This is an important contribution, but will it resolve the controversy of constant curvature vs constant area for CCP invagination? I doubt it. In some ways the controversy is somewhat contrived and, as this paper shows the answer is unlikely to be either or. Below are some specific comments, in somewhat random order, from someone (a curmudgeon?) who has reviewed and/or carefully read these papers since 1980. Points that the authors should address are in bold. All can be addressed with modifications to the text, as the one experiment I asked for (quantification of clathrin recruitment) is impossible with this approach).

1. I wonder how many people who cite Heuser's 1980 paper have ever read it carefully. Indeed, many of the observations made here were also made by Heuser. Below, for example, is a summary I wrote, but then removed from a review as it was too lengthy
"While Heuser favored the model that CCPs assemble first as flat structures and then rearrange during invagination, he was also careful to note several caveats. First, he observed that the edges of CCPs were 'ragged', likely reflecting sites of assembly of new polygons and that pentagons were more abundant at the edges. Thus, he argued that 'if even a few of these edge pentagons were destined to become completely surrounded with hexagons, it would be necessary to conclude that some degree of curvature can be built into coats as soon as they form'. Second, by examining tilted sections he observed that "even the flattest baskets have a small degree of inward curvature, and many were complete hemispheres". Finally, he cautioned that his images were snap-shots and a precursor-product relationship could not, therefore, be unambiguously established and that the very large flat lattices he observed might well be 'prove to be some sort of dead end'. We now know that fibroblasts, in particular, have large numbers of static flat clathrin plaques."

Thus, many of the author's conclusions, i.e. that 'completely flat clathrin coats are rare (pg 12, although they're not numbered), and that curved structures can be seen to emerge from the edges of flat lattices (see Supplemental Figure 1a, 3 examples on the right) are indeed consistent with Heuser's observations. In many ways, Heuser's 1980 paper is used as a straw man argument for the constant area model. The authors should more accurately cite and acknowledge this seminal paper.

We thank the reviewer for this insightful and constructive input on the interpretation of the constant area model (CAM). We have revised the discussion (**Page 14, Lines 397-402**), citing Heuser's observations more carefully and in similarity of what was already suggested eloquently by the reviewer. We agree that the strict interpretation of the CAM is misleading, and early evidence already suggests its flawed approximation of the endocytic mechanism (further mentioned now on **Page 15, Lines 429-431**).

2. As Heuser did in his 1980 classic, the authors here would do well to note several caveats related to their analyses. These include:
a. Like Heuser they have assembled static imaged to create a pseudotemporal model, albeit using a much more quantitative approach. Nonetheless, it seems that this assumes only a

single, stereotypic pathway for CCV formation. How good is this assumption? We know from dynamic imaging that there exists significant heterogeneity in both the kinetics and the molecular composition of CCPs. The authors should acknowledge this limitation.

We agree with the reviewer that the lack of direct temporal information is a clear limitation of our approach.

We now introduce this limitation on **Page 16, Lines 474-484**, where we discuss the disadvantage of reconstructing an average trajectory based on static images. Here, the assumption of a single, stereotypic pathway of endocytosis is addressed. We cannot exclude the possibility of slight mechanistic variations being averaged out using our approach. However, we want to highlight the fact that our approach seems sensitive enough to distinguish between structures that originate via endocytosis, and structures that derived from a different pathway, potentially from the Golgi.

We further address the kinetic variability in terms of abortive events on **Page 14, Lines 405-411**, and discuss their effect on the mechanistic interpretation of our results. Generally speaking, abortive events are characterized as dim and short-lived structures in live-cell acquisitions. As the earliest structures in our data set already contain half the final coat area, we are most likely not capturing these abortive events in the first place (potential technical reasons for not capturing earlier structures are discussed on **Page 14, Lines 385-395**).

b. The method, which required that they 'optimized the sample preparation to densely label clathrin at endocytic sites' involves labeling cells to near saturation with rabbit polyclonal antibodies to both clathrin light chains and clathrin heavy chains followed by detection with a second polyclonal donkey anti-rabbit. This gives 20 nm of additional and presumably flexible linker on the label. How might this effect the measurements and modeling? The Wu et al paper, which BTW has not been peer-reviewed, shows high precision fitting of the nuclear pore structure, but using endogenously tagged NUP-95, not two-layers of antibodies. The authors will need to discuss this limitation, it is my biggest concern regarding the analysis shown.

We acknowledge the limitations imposed by indirect immunolabelling and formulated a hypothesis on how this could affect our model fit (mentioned on **Page 13, Line 363**, illustrated in **Supplementary Figure 6**). A larger linkage error between label and target molecule would increase the distribution of localizations around the true underlying structure. As LocMoFit fits our spherical model directly to the localization coordinates, it is able to take this distribution into account, and will weigh the fit results based on the uncertainty of the localization estimation. A uniform distribution of labels around the true underlying structure should therefore be fitted accurately also at larger linkage error. A non-uniform labeling could occur should e.g. the densely crowded space between the coat and the plasma membrane not allow for the diffusion of the antibody to the clathrin epitopes. In that case, labeling would be one-sided, and instead of the true underlying structure, LocMoFit would optimize the spherical model to the highest probability density of label around + 10 nm from the true clathrin coat. This would result in an overestimation of the radius by the model, which we could correct by subtracting 10 nm from the experimentally determined radius. This was done in **Supplementary Figure 6** for the hypotheses of (1) uniform displacement by the antibodies; (2) biased displacement of the antibodies towards the cytosol; and (3) biased displacement of the antibodies towards the plasma membrane. Whilst we see that the fitting parameters scale with the corrected radii, the mechanistic interpretation of partial flat pre-assembly on the membrane, and subsequent bending and surface area growth still holds true.

3. One reason for continued controversy in this field is the lack of any attempt to resolve findings obtained using different methods. Can a parsimonious explanation be found, or are their artifacts or misinterpretations of previous findings that can explain the discrepancies? Any valid model should fit all of the valid data. For example, the authors fail to cite a recent paper by Willy et al in Dev Cell (PMID 34774130), which has been on BioRxiv since 2019 (doi: <https://doi.org/10.1101/715219>). Here, similar to this present study, the authors used high

resolution SIM-TIR to analyze ~1000 CCPs in 3 different cells lines (sadly non-overlapping with the cells used herein) and in *Drosophila* embryos to quantitatively test the two models. They conclude that their findings unambiguously support a constant curvature model. The authors would do the field a favor if they carefully read this paper and identified areas of commonality (i.e. that curvature is detected at early stages in both cases) and possible explanations for the discrepancies. Certainly, they should not ignore it.

We agree with the reviewer on the importance of consolidating findings from different studies to converge to a generally accepted mechanism of clathrin coat formation. We had indeed cited Willy et al in the introduction, but agree that further discussion of their findings should be included. We therefore discuss their findings in more detail, also in comparison to our work, on **Page 17, Lines 502-511**. We agree that we reach contradictory conclusions, which we think lies at least in part with the way that Willy et al. analyze their data. Willy et al. acquire 2D projections of the endocytic clathrin structures, whose size is just at the limit of their image resolution. They then compare their projected sizes to a purist constant area model, which assumes that a coat has to grow to its entire surface as an entirely flat structure and then instantaneously snaps to an increased curvature, resulting in a sudden drop of the projected area (footprint). As we and others (e.g. Bucher et al 2011, Heuser, 1980) have observed, completely flat lattices are rare, and curvature is initiated before final surface area is acquired. We do not agree that the absence of a purist constant area model implies that clathrin mediated endocytosis follows a constant curvature trajectory. Instead, we imagine that our cooperative curvature model is likely to fit well with the observations of Willy and colleagues.

4. An important body of evidence that is not considered in their model or discussion is that derived from live cell imaging. In addition to the heterogeneity mentioned above, studies have shown that the clathrin addition to CCPs is complete (i.e. the growth phase) occurs within the first ~20-30s, followed by a variable length (0->100s) plateau phase (Loerke et al, PMID 21447041). Both the current study and the Willy et al study admit that they may not be able to detect the earliest intermediates in CCP assembly. Indeed, in this study the smallest surface area CCPs are only 2-fold smaller than the largest CCPs, suggesting that over half of the triskelions have been recruited before a CCP can be distinguished from the background of clustered, nonspecifically-bound antibodies. Could the authors be monitoring events during the plateau phase and not the earliest events? Regardless, the findings are important as they address the nature of curvature generation during this plateau phase. While monitoring curvature generation during early events in CME, a recent study (Wang et al., eLife, PMID 32352376) showed that the acquisition of curvature within the first 20s of CCP assembly was a distinguishing feature between abortive and productive events. The authors might discuss how these studies on CCP dynamics might (or might not) inform their models.

We thank the reviewer for this very insightful comment and discuss this hypothesis on **Page 16-17, Lines 485-511**. We suggest that part of the initiating/growth phase observed in live-cell dynamics falls into the fast, flat assembly that we are unable to capture with our approach. It is challenging to clearly identify at which point in real-time we are detecting our earliest sites. We would however argue that the plateau phase in real-time could coincide with curvature generation and final addition of triskelia at the lattice rim. The variability in the duration of this plateau phase could therefore result from variable recruitment speed of triskelia and other factors during the finalizing of the vesicle neck.

5. The authors advertise 'quantitative' description of clathrin coated structure and indeed their measurements and models are quantitative; but there is no measure of intensity/numbers of triskelions and CCP growth: an important piece of quantitative data. I expect this is impossible with indirect immunofluorescence but should be considered as a limitation of the approach. Indeed, to my knowledge no one has yet quantitatively measured curvature generation in

parallel to clathrin addition at CCPs (closest is Saffarian and Kirchhausen, PMID 17993495), but they don't discuss the relationship.

We agree with the reviewer that quantifying the number of triskelia would be an essential piece of information to correlate area growth and curvature generation with dynamic information retrieved from fluorescence intensity in live-cell studies. Unfortunately, the indirect immunolabelling approach used in this work complicated this quantification, and direct comparison between number of localizations and fluorescence intensity cannot be made. However, we do observe a correlation between coat surface area and number of localizations in our data and show this in the newly added **Supplementary Figure 7**. This allows us to formulate the hypothesis on **Page 16-17, Lines 485-511**, which suggests that the plateauing of fluorescence intensity coincides with curvature generation and final triskelia addition to the coat rim. We further highlight the necessity of capturing both high spatial and temporal resolution simultaneously, to ultimately overcome this limitation.

6. On page 7 equation 1, you assume a constant growth rate for addition of triskelia, but later describe that the rate might be cooperative (as the number of edges increases). How would this affect your modeling?

We formulate the **surface area growth rate** of the clathrin coat to be proportional to the rim length with a **constant rate**. The cooperativity between clathrin molecules we consider to affect **the rate of curvature generation**. The more molecules are present, the more the entire coat is inclined to bent. We rephrased that section to emphasize this distinction (**Page 8, Line 217**).

Minor points:

- Can you indicate in the first paragraph of the results that you are using indirect immunofluorescence with rabbit anti-CLCA, anti-CHC and detection with donkey anti-rabbit for labeling, to augment the rather vague statement 'we optimized the sample preparation to densely label clathrin at endocytic sites'.

We added a clear indication on the labelling strategy used in this work on **Page 4, Lines 109-110**.

- I'm not comfortable with the conclusion on page 5 that your data 'indicates that at the time point of scission, the clathrin coat of nascent vesicles is still incomplete'. Other explanations might be the relative kinetics of scission vs CCP growth (i.e. these structures are too transient to detect), or that deeply invaginated pits are sheered-off the membrane during sample preparation (there is evidence that most biochemically isolated CCVs are derived from sheered CCPs).

We extended the explanation for the absence of fully closed vesicles with the hypotheses mentioned by the reviewer on **Page 5, Lines 159-161**.

- Bottom of page 5, can you briefly mention what data is shown in Supplemental Figure 2 (ie. Figure 2D and examples of likely non-endocytic CCPs shown in Supplemental Figure 2). When I read this, I questioned your speculation.

We clarified the cross reference to (now) Supplementary Figure 3 accordingly on **Page 6, Lines 184-185**.

- Can you indicate N CCPs from N cells in the data in Tables 2-3 for fibroblasts and U2OS cells? Do you observe and have to ignore a larger number of flat/clustered CCPs in the fibroblasts?

We indicated the number of cells and sites per data set in the Table captions on **Page 36, Lines 51; 959; and 967**. We did not quantify the number of flat/clustered, plaque like structures in our data sets. During data acquisition, we would specifically select cells with minimal number of these structures present, and

Full Revision

even within this cell chose an area in the periphery exhibiting low number of plaques. Our data is therefore not ideal to reliably quantify plaque density between different cell lines. Qualitative observations showed that whilst we had to disregard a few cells from the U2OS and SK-MEL-2 cell-lines due to high plaque formation, the 3T3 fibroblasts were relatively straight forward to image, as few cells showed high plaque density. A recent study by Hakanpää et al., 2022 (bioRxiv) showed the decreased formation of plaques when cells were seeded on fibronectin. The fact that fibroblasts excrete their own fibronectin agrees well with our observations of relatively few 3T3 cells exhibiting extensive plaque formation.

- The last 3 paragraphs of the Introduction are results. The Introduction might best be used to review literature in more detail, discuss the reasons why uncertainty still exists and perhaps indicate how the methods applied here will help.

We re-wrote the last 3 paragraphs of the introduction, now clearly stating the knowledge gap in the field, and what methods would be required to bridge it (**Page 3, Lines 80-102**).

Full Revision

Reviewer #1 (Significance (Required)):

This is another excellent addition to a growing list of papers seeking to define the process of curvature generation at endocytic clathrin coated pits. In my opinion, its impact would be increased by better integrating the results presented here with other studies and methods, including the recent paper by Willy et al and the large body of literature on coated pit dynamics, some of which might be relevant in interpreting results, or at least placing them in a real vs pseudo-temporal perspective. The methods introduced and the quality of imaging, modeling and quantification further increase the study's significance. The finds will be of interest to those in the CME field, those studying membrane curvature generation in other contexts, those modeling CME, vesicle formation and curvature generation and those using SMLM to discern the structure of macromolecular assemblies.

Reviewer expertise: Clathrin-mediated endocytosis (Sandra Schmid)

Reviewer #2 (Evidence, reproducibility and clarity (Required)):

Summary

In this article, the authors aimed to investigate the dynamic of clathrin lattice during clathrin-mediated endocytosis (CME). Overall, they successfully achieved the goal by observing a large number of clathrin spots from several cell lines with 3D single-molecule localization microscopy (SMLM). With the help of this high-resolution imaging technique, they were able to describe the physical properties of each spot and reconstruct the assembly and remodeling of the clathrin coat. Moreover, by comparing the constant area/curvature model with their own data, the authors highlighted that neither of the prevailing models perfectly explained what they observed and proposed 'cooperative curvature model'. With the novel model, the authors were able to reconstruct the clathrin coat remodeling in different cell lines and concluded that the simultaneously bending and assembly of the clathrin coat is a homogenous property of endocytosis.

The experiments and analytical procedures are well-designed and performed, and the manuscript is well-organized. The conclusion 'cooperative curvature model' was deduced from a large amount of data analysis and clearly stated in the text. I would like to recommend its publication if the following issues will be clarified.

Major comments:

1. The authors compared the morphological dynamics of clathrin-coated pit among three different cell lines (SK-MEL-2, U2OS, and 3T3) and found slight differences. As U2OS cells was derived from bone tissues, it has different mechanical properties (membrane tension, elasticity of cortical layer, etc..). It would be interesting to consider those mechanical properties in understanding the morphology (Figure 2) and progress (Figure 4) of the CME. Considering the fact that the bending energy of the plasma membrane is dependent on the membrane tension, they may be able to find some relationships between mechanical properties of the cell cortex and CME.

We thank the reviewer for this comment and very much agree that the relationship between mechanical properties structural adaptation of the endocytic machinery is a highly interesting question. We came to the same conclusion and are therefore exploring this relationship at the moment. This is however not a straightforward task, and the complex nature of plasma membrane mechanics necessitates careful experimental design. It is therefore outside the scope of this publication. We do think this point further highlights the potential of the method presented here, as it allows the investigation of additional principles in clathrin-mediated endocytosis mechanics. We do hope to share our insights on this topic soon.

2. In Figure 4, the authors estimated the progression of the CME using the frequency distribution of theta. However, I wonder how they handled the events which were aborted in the middle of the CME. It had been suggested that some CME are aborted during the initial step of the CME. The authors should consider (at least discuss) those abortive events, which can disturb the analysis.

Generally speaking, abortive events (now discussed on **Page 14, Lines 405-411**) are characterized as dim and short-lived structures in live-cell acquisitions. As the earliest structures in our data set already contain half the final coat area, we are most likely not capturing these abortive events in the first place (potential technical reasons for not capturing earlier structures are discussed on **Page 14, Lines 385-395**).

Abortive events throughout the later process of endocytosis would, according to our data, still follow the same mechanistic trajectory as other sites. They could potentially slightly skew our pseudotime analysis, as they would result in an overestimation of specific endocytic stages. The overall mechanistic insight of our work would not be greatly affected, as curvature generation would still occur according to the same trajectory. Due to the low impact on our overall results we do not discuss these late abortive events further.

Minor comments:

1. Page5, result section 2. The author should further explain why vesicles from trans Golgi could be responsible for the small disconnected set of data points corresponding to the vesicles with larger curvatures.

We extended our explanation for the presence of non-endocytically derived structures in our data set on **Page 6, Lines 184-189**. We further extended the supplementary information with an additional experiment (**Supplementary Figure 4**), highlighting the absence of AP2-positive structures within the disconnected population. As AP2 is a specific marker for CME, these results further solidify our hypothesis. Further experiments would be required to determine their exact origin, and are outside of the scope of this publication.

2. Page7, line 6. The author assumed that the clathrin coat starts growing on a flat membrane. However, as is mentioned in the discussion, clathrin has been proved to have curvature sensing ability which could be further amplified by adapter proteins by several times (Zeno et al., 2021). So, it seems that clathrin preferred a highly curved membrane instead of a flat one. Is it still reasonable to make this assumption?

Whilst our assumption states the growing of clathrin coat on flat membranes, we do not restrict our model to an intercept through 0, and it would therefore still hold true even in the case of growth starting on slightly bent membranes. The impact of the preference of clathrin for curvature is considered as a potential mechanistic explanation for the positive feedback in curvature generation described by our model. We therefore already cite the reference mentioned by the reviewer on **Page 8, Line 224**.

As we do observe flat structures in our data set (discussed more in detail now on **Page 14, Lines 396-404**), we still think the assumption of early flat growth holds true.

3. Page 9, result section 4. In the sentence: "we effectively generated the average trajectories of how curvature, surface area, projected area and lattice edge change during endocytosis in SK-MEL-2 cells (Figure 4B-E)." Here I think the authors are describing Figure 4C-F.

That is correct, an oversight on our part. We changed the cross-reference.

4. Page 11, discussion. In the sentence: "A deviation of the cross-sectional profile from a circle is nevertheless preserved in the averaging (Supplementary Figure 5)." I didn't see supplementary figure 5 in the article.

We changed the cross-reference. We were addressing a subsection of Supplementary Figure 8.

Reviewer #2 (Significance (Required)):

From a vast amount of microscopic images and data analysis, the manuscript gives a clear model on the progress of the CME, which integrates two opposing models; constant area and constant curvature models. This is a big progress in our understanding of the molecular mechanism of CME, and will attract many researchers in the field of cell biology. From a viewpoint of my expertise (molecular imaging of plasma membrane and endocytic processes), this manuscript has significant impact on the related research fields.

Reviewer #3 (Evidence, reproducibility and clarity (Required)):

Summary:

The authors used single-molecule localization microscopy of clathrin in fixed cells (2 human cell lines, one mouse) to capture snapshots of a clathrin-mediated endocytosis (CME), fitted these localizations to a geometric model of a forming vesicle, and used these fitted measurements to test existing models of clathrin-mediated vesicle formation before refining their own. Specifically, the closing angle, a measure of vesicle completeness, was used as a proxy for growth-stage of the vesicle such that the many captured snapshots could reconstruct a pseudo-timeline with an unknown parameterization of time on closing angle. Two standard models of CME vesicle formation, where the surface area is kept constant or where the curvature is kept constant, were examined and determined to be incommensurate with the pseudo timelines of curvatures and surface area. The authors then describe their own model for CME vesicle formation, in which neither surface area nor curvature are constant in evolution of the vesicle, and cooperative forces are hypothesized to non-linearly modulate the curvature-growth as a function of closing angle. Additionally, by binning snapshots and then aligning, scaling, and azimuthally smoothing each bin, they reconstruct representations of distinct endocytic stages.

Major comments:

Most results are quite convincing, and the authors do a nice job of displaying examples of SMLM data, both with fit results as well as example clathrin assemblies that are too far removed from their budding-vesicle model to be included for analysis, for example. It is also worth noting that the clathrin images themselves appear to be very high-quality - clearly, as detailed in the methods, attention was given to each step of the imaging and reconstruction process. While the presented cooperative curvature model seems reasonable and surely fits the curvature-, surface area-, and rim length-vs. closing angle data better than the simplistic constant surface-area and constant curvature models, it also has more parameters, namely: γ (the initial rate of curvature change with closing angle) and H_0 (the final preferred curvature). It would be appropriate to calculate an information criterion (e.g. Bayesian), using an assumption of Gaussian-distributed errors (presumably the data fitting in R was least squares, so this would match) to justify the additional parameters.

This is an important observation by the reviewer. Indeed, our model uses one more parameter compared to the models we compare it with. To justify this, we performed the calculation as suggested by the reviewer, and found that the cooperative curvature model (CoopCM) indeed results in the lowest BIC (**Supplementary Notes**). We therefore are confident that out of the three models tested in this work, our CoopCM fits best to the underlying experimental data (**Page 8, Lines 232-235**)

A related issue relates to the error in the extracted value of the closing angle from a single 3D reconstruction - the error distribution should be quantified for this very important parameter. The errors in the other parameters extracted from the fits are less important, but would enhance the paper.

We thank the reviewer for pointing out the importance of the estimation error of the key parameter closing angle. To address this point, based on the geometrical model, we simulated clathrin-coated structures with closing angles evenly distributed across the entire range (0-180°). This realistic simulation represents the data quality (e.g., localization precision and labeling efficiency) of the experimental data (corresponding methods are included in **Pages 22- 23, Lines 679-706**). The result of fitting these structures using LocMoFit shows an unbiased estimation with small spread of the error (overall STD = 2.82°; see the newly included **Supplementary Figure 2a**).

Full Revision

Pseudo-temporal sorting on closing angle makes sense and I appreciate the authors mentioning potential caveats to the monotonicity, etc. However, a comment about the impact of closing angle errors on the pseudo-time determinations would be helpful. The agreement of theta-rank plots with the hypothesized \sqrt{t} scaling is reassuring.

I additionally appreciate the robustness of fitting a geometric structure from localizations rather than relying on pseudo-temporal sorting on clathrin count extracted from localization-merging of multi-blinking emitters.

The pseudo-temporal sorting is based on the precisely estimated closing angle, and therefore is also precise, as the distribution of the fitted closing angle has no significant distortion compared to the expectation (**Supplementary Figure 2b**).

The authors did a nice job of qualifying their more speculative claims, in particular I appreciated their mentioning the possibility that smaller clathrin coats could be below their detection limit. The authors state a set of data points in suppl. figure 2D (and suppl. Fig 3A-C) are "likely" small clathrin-coated vesicles from the trans Golgi. I appreciate the examples rendered in that figure so a reader can appraise, but if they have my background they might not know how reasonable exclusion of this data is from model testing. This claim could be rephrased or the rationale expanded upon to justify the Golgi hypothesis.

We agree with the reviewer and further expanded on our hypothesis on the origin of the structures within the disconnected cloud of data points (**Page 6, Lines 184-189**). We further performed an additional experiment (**Supplementary Figure 4**), where we simultaneously imaged the clathrin coat at high resolution, and the CME specific AP2 complex tagged with GFP at diffraction limited resolution. We observed that there were no AP2-GFP positive structures present in the disconnected cloud of our data set, and conclude that these structures indeed must originate via a different pathway.

The data and methods are presented such that they could be reproduced, and replicating their experiment in multiple cell lines, across multiple species, would seem to be adequate replication. As mentioned above, the statistical analysis of whether the model complexity is justified by improved goodness of fit is currently missing but can readily be checked and added. Minor comments:

Last paragraph of the introduction, positive feedback is mentioned but not the slowing down as preferred curvature is realized (inclusion of which might help foster a clearer understanding of the model early on).

We now mention the slowing down towards a preferred curvature in our introduction on **Page 3, Lines 100-102**.

In Fig. 1, please state in the figure caption what is being displayed in the two large panels and what is the color map. Is this the 3D data from the overlapping elliptical Gaussians projected on the plane in a "hot" map? Further, in the top right small panels, are the x-y images projections of all z, or measured at a specific z?

We adjusted Figure 1 and the figure caption to clearly explain what is mentioned in each superresolution panel. The exact details for image rendering, including the color map and gaussian blurring of the localization coordinates are now described in the methods on **Page 21, Lines 625-627**. Ultimately, the x-y images represent an enlarged view of the projections as visible in the previous two panels. We hope that rephrasing of Figure 1 legend clarifies this accordingly.

In Eqn. (1), epsilon is not defined.

Full Revision

The definition is mentioned on **Page 8, Line 210**, right before the equation, same as for k_{on} .

For the theta-rank plots (Fig4 B, SFig D-F ii) moving the $\theta(t)=\sqrt{t}$ red curves behind sorted theta data would make the data easier to see.

We adjusted the Figures according to the reviewer's suggestion.

"Laser" in sentence about the speckle reducer should probably be plural.

We corrected this grammar mistake, and changed "laser" to "lasers" on **Page 20, Line 586**.

I would like to see the "custom" algorithm based on redundant cross-correlation for drift correction briefly described.

We added an explanation on the algorithm used for the drift correction on **Pages 20-21, Lines 611-617**.

A legend for supplemental figure 3 A-C would be nice.

We added a legend for the various models in (now) **Supplementary Figure 5**, and further made some clarifications in the figure caption.

If the definition of the abbreviation flat-to-curved-transition as FTC was explicit I missed it.

As we do not use this abbreviation anywhere else in the manuscript, we removed it from the **Supplementary Note** to avoid confusion.

Resolution of 20 and 30 nm (laterally and axially, respectively) was quoted once towards the beginning of the manuscript as being an improvement resulting from the localization method described in Li et al., 2018. Resolution can be difficult to speak about precisely, but the methods section would seem to indicate that localizations are filtered at 20 nm lateral localization precision (potentially 30 nm axially?), and I think the authors could consider rephrasing to depict this unless I am missing elsewhere a description of the resolution metric being used.

The original 20 and 30 nm resolution (laterally and axially) was calculated based on the median localization precision values in x-y and z for a representative image, using the FWHM approach (described in Methods **Page 21, Lines 621-624**). After consideration of the reviewer's question, we found the modal value to be a better quantity to calculate the resolution, and changed this in the text accordingly (**Page 4, Lines 113-115, and Methods Page 21, Lines 621-624**).

Reviewer #3 (Significance (Required)):

Proteins involved with inducing curvature in membranes are in general very exciting targets for localization microscopy, yet still for many systems questions remain unanswered. The authors tackle one such question in this manuscript. In other, unresolved, discussions, the posed hypotheses are quite similar to the simplistic models surpassed in this work (e.g. that curvature scales linearly with local protein copy number, or that surface area scales linearly with local protein copy number). The idea of cooperativity may be useful for others to consider, and the authors additionally demonstrate a seemingly smooth workflow using their separately described tools (primarily LoMoFit; Wu et al. 2021).

I myself am not an expert on CME or vesicle trafficking. My background is primarily in SMLM method development and SMLM / fluorescence image analysis. From my perspective, the

novelty of the biological conclusions appears to be the authors' specific cooperative model and the presence of two structural states which are enriched (closing angle 70° and 130°). As referenced, and authors F. Frey and U. S. Schwarz nicely present in Bucher et al. 2018, the constant curvature and constant surface area models are known to be inaccurate descriptions of CME evolution, and further it is also known that clathrin first assembles small flat structures before beginning to curve the membrane. However, the 3D super-resolution imaging and direct evaluation of a 3D model geometry in this work is a nice extension of the 2D super-resolution imaging and projection evaluation in the authors' previous work studying endocytosis through ensemble averaging in yeast (Mund et al. 2018) as well as the analysis on projections in Bucher et al. 2018. Fully 3D treatment of the clathrin structures allows the authors to orient asymmetric assemblies such that they are averaged out in their ensemble reconstruction, and as they point out the molecular specificity afforded by a fluorescence-based technique ensures unbiased segmentation of clathrin-involved endocytic sites. In other words, while this work does not describe a technical advance not already described elsewhere, it sets a nice example for those researching protein-membrane interactions of how to leverage the right tools to clearly and directly answer their questions. With their additional work to make these tools extensible to other geometries, multiple color channels, etc., I expect their work to inspire quality studies in other systems. That significance is complementary to their proposal of a reasonable model for the geometric evolution of CME.

References:

Maximum-likelihood model fitting for quantitative analysis of SMLM data, Yu-Le Wu, Philipp Hoess, Aline Tschanz, Ulf Matti, Markus Mund, Jonas Ries, bioRxiv 2021.08.30.456756; doi: <https://doi.org/10.1101/2021.08.30.456756>

Bucher, D., Frey, F., Sochacki, K.A. et al. Clathrin-adaptor ratio and membrane tension regulate the flat-to-curved transition of the clathrin coat during endocytosis. Nat Commun 9, 1109 (2018). <https://doi.org/10.1038/s41467-018-03533-0>

Markus Mund, Johannes Albertus van der Beek, Joran Deschamps, Serge Dmitrieff, Philipp Hoess, Joeske Louise Monster, Andrea Picco, François Nédélec, Marko Kaksonen, Jonas Ries,

Systematic Nanoscale Analysis of Endocytosis Links Efficient Vesicle Formation to Patterned Actin Nucleation, Cell, 174, 4, (2018). <https://doi.org/10.1016/j.cell.2018.06.032>.

s

July 6, 2022

RE: JCB Manuscript #202206038T

Dr. Jonas Ries
European Molecular Biology Laboratory
Meyerhofstr. 1
Heidelberg 69117
Germany

Dear Dr. Ries:

Thank you for submitting your revised manuscript entitled "Superresolution microscopy reveals partial preassembly and subsequent bending of the clathrin coat during endocytosis". We would be happy to publish your paper in JCB pending final revisions necessary to meet our formatting guidelines (see details below). As previously discussed, our policy is that to be published in JCB all methods being used in a manuscript must be published in a peer-reviewed journal. Therefore, publication is also contingent upon the preprint "Wu, Y.-L., Hoess, P., Tschanz, A., Matti, U., Mund, M., and Ries, J. (2021). Maximum-likelihood model fitting for quantitative analysis of SMLM data. *BioRxiv* 2021.08.30.456756. <https://doi.org/10/gmnmvqv>" being formally accepted.

A. MANUSCRIPT ORGANIZATION AND FORMATTING:

- 1) Text limits: Character count for Articles is < 40,000, not including spaces. Count includes abstract, introduction, results, discussion, and acknowledgments. Count does not include title page, figure legends, materials and methods, references, tables, or supplemental legends.
- 2) Figures limits: Articles may have up to 10 main text figures.
- 3) Figure formatting: Scale bars must be present on all microscopy images, including inset magnifications. Molecular weight or nucleic acid size markers must be included on all gel electrophoresis.
- 4) Statistical analysis: Error bars on graphic representations of numerical data must be clearly described in the figure legend. The number of independent data points (n) represented in a graph must be indicated in the legend. Statistical methods should be explained in full in the materials and methods. For figures presenting pooled data the statistical measure should be defined in the figure legends. Please also be sure to indicate the statistical tests used in each of your experiments (either in the figure legend itself or in a separate methods section) as well as the parameters of the test (for example, if you ran a t-test, please indicate if it was one- or two-sided, etc.). Also, if you used parametric tests, please indicate if the data distribution was tested for normality (and if so, how). If not, you must state something to the effect that "Data distribution was assumed to be normal but this was not formally tested."
- 5) Abstract and title: The abstract should be no longer than 160 words and should communicate the significance of the paper for a general audience. The title should be less than 100 characters including spaces. Make the title concise but accessible to a general readership. *Please consider a title that conveys the novel model being proposed. We also discourage references to the specific techniques in the title, therefore "Superresolution microscopy" should be removed. *
- 6) Materials and methods: Should be comprehensive and not simply reference a previous publication for details on how an experiment was performed. Please provide full descriptions in the text for readers who may not have access to referenced manuscripts.
- 7) Please be sure to provide the sequences for all of your primers/oligos and RNAi constructs in the materials and methods. You must also indicate in the methods the source, species, and catalog numbers (where appropriate) for all of your antibodies. Please also indicate the acquisition and quantification methods for immunoblotting/western blots.
- 8) Microscope image acquisition: The following information must be provided about the acquisition and processing of images:
 - a. Make and model of microscope
 - b. Type, magnification, and numerical aperture of the objective lenses

- c. Temperature
- d. Imaging medium
- e. Fluorochromes
- f. Camera make and model
- g. Acquisition software
- h. Any software used for image processing subsequent to data acquisition. Please include details and types of operations involved (e.g., type of deconvolution, 3D reconstitutions, surface or volume rendering, gamma adjustments, etc.).

9) References: There is no limit to the number of references cited in a manuscript. References should be cited parenthetically in the text by author and year of publication. Abbreviate the names of journals according to PubMed. *Please note our policy for citing preprints, such as that from Jin et al 2021; Kaplan et al 2022; Sochacki et al 2020. <https://rupress.org/jcb/pages/reference-guidelines> Supplemental numbered references are also not permitted.

10)* Supplemental materials: There are strict limits on the allowable amount of supplemental data. Articles may have up to 5 supplemental figures, therefore please move some of your supplemental figures to the main text and be sure to correct the callouts accordingly. Please also note that tables, like figures, should be provided as individual, editable files. A summary of all supplemental material should appear at the end of the Materials and methods section. *

13) ORCID IDs: ORCID IDs are unique identifiers allowing researchers to create a record of their various scholarly contributions in a single place. At resubmission of your final files, please consider providing an ORCID ID for as many contributing authors as possible.

Please note that JCB now requires authors to submit Source Data used to generate figures containing gels and Western blots with all revised manuscripts. This Source Data consists of fully uncropped and unprocessed images for each gel/blot displayed in the main and supplemental figures. Since your paper includes cropped gel and/or blot images, please be sure to provide one Source Data file for each figure that contains gels and/or blots along with your revised manuscript files. File names for Source Data figures should be alphanumeric without any spaces or special characters (i.e., SourceDataF#, where F# refers to the associated main figure number or SourceDataFS# for those associated with Supplementary figures). The lanes of the gels/blots should be labeled as they are in the associated figure, the place where cropping was applied should be marked (with a box), and molecular weight/size standards should be labeled wherever possible.

B. FINAL FILES:

****It is JCB policy that if requested, original data images must be made available to the editors. Failure to provide original images upon request will result in unavoidable delays in publication. Please ensure that you have access to all original data images prior to final submission.****

****The license to publish form must be signed before your manuscript can be sent to production. A link to the electronic license to publish form will be sent to the corresponding author only. Please take a moment to check your funder requirements before choosing the appropriate license.****

Thank you for this interesting contribution, we look forward to publishing your paper in Journal of Cell Biology.

Sincerely,

Pier Paolo Di Fiore, MD, PhD
Editor

Andrea L. Marat, PhD
Senior Scientific Editor

Journal of Cell Biology

Reviewer #1 (Comments to the Authors (Required)):

I originally reviewed this paper for Review Commons. The authors have addressed my concerns and the paper is more balanced. As I originally stated this remains a well-executed and technically advanced study of curvature generation during CCV formation. The CoopCM model is much more attractive than the more extreme interpretations of the constant curvature and constant area models.

Reviewer #2 (Comments to the Authors (Required)):

I am happy to see the Bayesian Information Criterion added to this work and that the metric in support of the authors' proposed model across all cell types. Further, the detailed SMLM simulation to determine error distribution on fitted closing angle supports that this critical parameter is robustly quantified. Finally, the authors conducted an additional experiment which supports their hypothesis that a distinct subset high-curvature clathrin structures are not a part of the CME pathway and can therefore be excluded from their analysis. Having then addressed each of my major comments, I recommend this manuscript for publication in the Journal of Cell Biology.

Minor issues:

- Line 296 contains the grammatical error 'A fast transitions'.
- The addition of SMLM rendering details is much appreciated. One final point that Gaussian 'width' can often be unclear. Is the rendered 3 nm width sigma, FWHM, $1/e^2$?